# Sanitizing LLMs: Retrospective Learning for Self-Correction of Inconsistent Samples via User Preferences

## Abstract

With the advent of large language models (LLMs), using LLMs in conjunction with prompt-based tasks has demonstrated the ability to reduce the high cost and inefficiency of human annotations. Nonetheless, in unsupervised new downstream tasks that require user preferences to align data annotations with expectations, existing evaluation methods for prompt-based tasks become ineffective, especially when ground truth annotations are insufficient or missing. To fill this gap, we propose the novel Consistent and Inconsistent (CAI) Ratio, inspired by our experimental observation that LLMs underperform when the number of inconsistent samples—those with inconsistent predictions across LLMs and the student model—exceeds the number of consistent samples. By estimating the CAI ratio and identifying consistent and inconsistent samples with our proposed CAI identification approach, we aim to minimize inconsistency and enhance the accuracy of LLM-generated annotations for unsupervised data. To achieve this, we introduce Retrospective Learning (*RetroL*) with user preference, a data-centric approach that collaborates with the student model and LLMs, using a small number of human annotations as user preferences to resolve inconsistencies in the identified samples. Applied to eight domain-specific NLP datasets, our Retrospective Learning approach, leveraging CAI identification, significantly improved the accuracy of LLM-generated responses, with the CAI ratio increasing as the accuracy improved.

## 1 Introduction

Large language models (LLMs), with their unprecedented zero-shot performance, as shown by Kojima et al. (2022), have seen burgeoning deployment across various domains of NLP problems. In particular, LLMs are being leveraged as teachers, alongside smaller pre-trained models as student learning paradigms, to generate annotations and mitigate the inefficiencies, high costs, and dependence on notoriously laborious manual annotation (Chen et al., 2024). However, it has been demonstrated that LLMs possess intrinsic drawbacks, such as randomness, inconsistency, as noted by Sclar et al. (2024) and Atreja et al. (2024), and hallucination, which can detrimentally impact the trustworthiness of their generated output. To address these issues, prompt-based learning tasks have emerged. Several studies, including Brown et al. (2020) and Chen & Tsang (2024), explore these tasks, along with other research (Wei et al., 2021; Yao et al., 2022; Diao et al., 2023; Liu et al., 2023; Wang et al., 2023; Wei et al., 2022; Yao et al., 2024; Long, 2023; Huang et al., 2022; Madaan et al., 2024; Huang et al., 2023; Shinn et al., 2024). Devising specific and effective evaluation metrics for LLMs is vital to improving LLMs' performance across various prompt-based tasks. Conventionally, many of these tasks rely on ground truth annotations from the training dataset to evaluate proposed prompts. Feedback from this evaluation is then utilised to iteratively refine the prompts, by improving LLMs performance on the testing dataset. However, in unsupervised downstream tasks with user preferences, in which explicit guidance is not in provision, it becomes crucial to design a learning process that encourages annotations to align with user preferences to improve the quality of training for a new downstream model. This challenge is commonly encountered and ubiquitous in many real-world applications, especially in intent classification, sentiment analysis, and recommendation systems, where the user or expert preference alignment is essential for generating satisfying annotations. For instance, in AI chatbots like ChatGPT, unsupervised data—queries or questions from users with the same intention

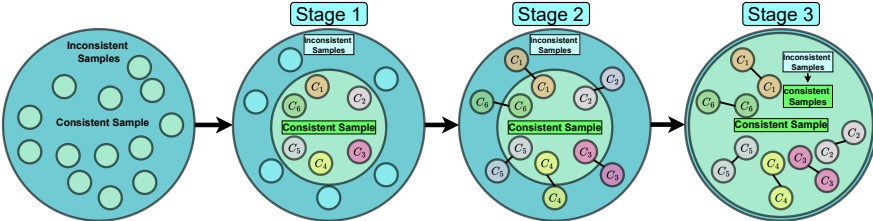

Figure 1: **Schematic Depiction of Retrospective Learning (RetroL)**. The inner circle is the consistent sample set $\mathcal{C}$, and its expansion (the outer circle) is the set that covers all the inconsistent samples $\mathcal{I}$. The third circle shows our proposed *Sanitizing LLMs* solving inconsistent (difficult) sample issues. The $C$ indicates each category of the samples. The line connects small circles in the out circle and inner circle, symbolising the cosine similarity between the inconsistent sample and consistent sample, the highest ones assigned to the corresponding class (**See Section 3**).

but expressed in different formats or languages—without predefined user preference categories can result in responses that fail to meet user expectations. Thus, a user preference-based annotation process and evaluation is essential to ensure that model-generated responses appropriately align with end-user needs. While it might seem intuitive to use LLMs alone for annotation, they may struggle with data that involve specific user preferences for a new downstream task. Furthermore, in unsupervised learning tasks that rely on user preferences to align data annotations with expectations, where the competency of the teacher model (LLMs) is uncertain, and no external knowledge is provided, evaluating annotations generated by the LLM becomes a significant challenge. As Zhou et al. (2024) demonstrate, these annotations are often prone to overconfidence in their predictions, which necessitates the implementation of self-supervised mechanisms for self-correction (Xiong et al., 2023). In this scenario, relying solely on a student model for fine-tuning or training from scratch is also not feasible, given the lack of supervision. In summary, there are two critical challenges in new downstream tasks with only an unsupervised dataset and user preferences:

- **Challenge on Evaluation:** *How can we evaluate the performance of LLMs or student-generated annotations based on user preferences when dealing with unsupervised data?*

- **Self-Correction for Unsupervised Tasks with Limited User Preferences:** *Given an unsupervised task that lacks annotations for fine-tuning LLMs and training a student model with only a small set of user preferences, how can we enable self-correction to improve annotation accuracy for both models without relying on any external knowledge?*

To address the challenge of evaluation, **Consistent and Inconsistent (CAI)** ratio (**see Section 3.2**), the first evaluation metric designed for unsupervised textual datasets in prompt-based tasks. Our experimental study reveals that LLMs tend to perform poorly when the number of consistent samples with consistent predictions between LLMs and the student model outnumbers the inconsistent samples. While the CAI ratio can partially assess the performance of LLMs and the student model on a given unsupervised dataset, it does not fully resolve the issue of inconsistent outputs, as evidenced by the identified inconsistent samples. These samples represent a subset of the training data that has demonstrated inconsistent predictions across both LLMs and the student model, with significantly lower annotation accuracy (**see Figure 3**).

Furthermore, if the incorrect annotations of the identified inconsistent samples can not be self-corrected, this incorrect annotation will pass to the student model, resulting in poor generalisation. Thus, identifying and being able to self-correct inconsistent samples are crucial for enhancing the consistency and accuracy of LLM-generated annotations. To address this challenge, ***Retrospective Learning (RetroL)*** is proposed. RetroL uses a divide-and-conquer self-correction (DCSC) technique in conjunction with the identified consistent samples, which are identified with much higher accuracy than the inconsistent samples. The inconsistent samples are self-corrected via the DCSC process, which employs a top-nearest embedding scheme and majority voting. By utilising the CAI identification and DCSC, our RetroL consistently increased classification accuracy with a higher CAI ratio when we applied it to eight domain-specific datasets.

## 2 BACKGROUND

### 2.0.1 LLMs FOR DATA ANNOTATION

LLMs have exhibited pre-eminent competency in dealing with text annotation tasks for many open-domain tasks, such as open-domain spoken language understanding (Chen et al., 2024; 2023), and frequently outperform crowdsourcing and manual annotation without requiring training on specific data (Gilardi et al., 2023). However, the development of robust evaluation metrics and effective approaches for adapting LLMs to unsupervised textual data with user-defined preferences remains an open challenge. Our work fills the gap.

### 2.0.2 LLM AND STUDENT ANNOTATION PARADIGMS

Previous works, as highlighted by Thapa et al. (2023), emphasize the importance of teacher-student models in achieving superior performance (Chen et al., 2024; 2023). Recently, Gligorić et al. (2024) have proposed collaborating between LLMs and human annotation to search for unbiased, accurate annotations and for an optimal balance between the high cost of human annotation and LLMs' affordability and efficiency. However, it does not solve the vital aspects of evaluation and self-correction in our problem setting. Simply using a student model with a specifically designed loss function cannot solve this issue effectively. Moreover, the previous student and teacher paradigm overlooks exploiting inconsistent (difficult) samples, an integral aspect contributing to the degradation of model performance. To the best of our knowledge, no work has explored collaboration between LLMs and student models with a small number of human annotations for self-correction for the inconsistent sample based on our proposed CAI ratio (See Section 3).

### 2.1 PROBLEM SETTING

Given unsupervised text corpus distributions for testing and training, denoted as $\mathcal{D}_u = \{x_1, \ldots, x_N\}$ and $\mathcal{D}_T = \{x_1, \ldots, x_L\}$, where $x \in \mathcal{X} \subseteq \mathbb{R}^d$. A set of user-preference samples $H$ is also given. These user-preference samples are clustered into $k$ clusters $C_1, C_1, ..., C_k$. In addition, a set of preference annotations is also given, denoted as $\mathcal{Y} = \{\bar{y}_1, \bar{y}_2, ..., \bar{y}_k\}$. Each $C_j$ is denoted as $C_j = \{(x_i, \bar{y}_j) | x_i \in H_j\}$ and $H_j \subseteq H$. $H = \{(x_i, \bar{y}_i)\}_{i=1}^s$, with $s = 5\%$ of $|\mathcal{D}_T|$. Each Cluster does not overlap, such that $(C_i \cap C_j = \emptyset, \forall i \neq j)$, and the union of all clusters covers $H$. These user-preference samples incorporate user preferences for alignment purposes. The learning objective is to assign a user preference label $\bar{y} \in \mathcal{Y} = \{1, \ldots, k\}$ correctly to each $x$. We assume that the distribution $\mathcal{D}_u$ can be partitioned into two subsets: consistent samples $\mathcal{C}$ and inconsistent samples $\mathcal{I}$, such that $\mathcal{C}, \mathcal{I} \subseteq \mathcal{D}_u, \mathcal{C} \cap \mathcal{I} = \emptyset$, and $|\mathcal{C}| + |\mathcal{I}| = |\mathcal{D}_u|$. However, in practice, the consistent and inconsistent subsets are not known in advance and must be estimated **(see Section 3.2)**. The consistent and inconsistent samples are identified using the student model $\mathcal{S}$ and teacher model $\mathcal{T}$, along with a small set of user-preference samples. The learning objective is to minimize the inconsistency (i.e., reduce the size of $\mathcal{I}$) and maximize the annotation accuracy of the LLMs on $\mathcal{D}_u$.

## 3 SANITIZING LLMs PARADIGAM

### 3.1 PROCUREMENT OF ANNOTATED SAMPLE DISTRIBUTIONS FROM STUDENT AND TEACHER MODELS

**Annotation Assignment Using Student Model:** The initial step is to align each instance of unsupervised data with annotations according to user preferences. We start by using MINILM Wang et al. (2020), a sentence-transformers model, as our student model denoted as $\mathcal{S}$, to first acquire $\mathcal{S}(x_i) = e_i$, sentence embeddings, for each $x_i$. Thereafter, we apply our proposed *user preference-based majority voting* approach inspired by Mostafazadeh Davani et al. (2022) to assign annotations based on our proposed Average Similarity(AS) function as follows:

$$AS(e_i, C_j) = \frac{1}{k} \sum_{e \in \text{Top-}k(C_j, e_i)} \frac{e_i \cdot e}{\|e_i\| \|e\|}, \tag{1}$$

where $e_i$ denotes the embedding for $x_i$, and $e$ represents the embedding of each sample in cluster $C_j$. The term Top-$k(C_j, e_i)$ refers to the subset of samples in $C_j$ with the top $k$ cosine similarity

scores with $e_i$. Formally, Top-$k(C_j, e_i) = \{e \in C_j \mid \text{AS}(e_i, e) \text{ is among the top } k \text{ in } C_j\}$. Based on the calculated cosine similarity, the examples most similar to $e_i$ are identified, and the average cosine similarity is computed for the top-selected samples in each cluster. In our experiments, we set $k$ to five. Lastly, for the annotation assignment, we assign the label of the cluster $C_j$ with the highest average cosine similarity score to the unlabelled sample $x_i \in D_u$. The cluster $C_{j*}$, which has the highest average cosine similarity with the embedding $e_i$ of a sample $x_i$, is defined as:

$$C_{j*} = \arg\max_{C_j} \text{AS}(e_i, C_j) \tag{2}$$

where $\text{AS}(e_i, C_j)$ is the average cosine similarity of $e_i$ with the embeddings in $C_j$. The annotation $\bar{y}_{j*}$ associated with $C_{j*}$ is then assigned to $x_i$, i.e., $\bar{y}_i = \bar{y}_{j*}$. This process is represented by the annotation assignment function $h(x_i)$. Subsequently, the annotation associated with $C_{j*}$, as defined by the user, will be assigned to $x_i$. Finally, the student-annotated dataset $D_s = \{(x_i, \bar{y}_i)\}_i^N$, where each $\bar{y}_i$ represents the user preference-based annotation, is obtained by following the user preference-based majority voting annotation approach.

**Annotation Assignment Using Teacher Model(LLMs):** With the acquired dataset $D_s = \{(x_i, \bar{y}_i)\}_{i=1}^N$, we further exploit LLMs using zero-shot prompting (without including annotations from the student) and single-shot prompting (including annotations from the student) through a group prompting method to provide annotations for each $x_i$. We define the annotations as $\bar{y}_i^t = T(x_i)$ for zero-shot prompting and $\hat{y}_i^t = T(x_i, \bar{y}_i)$ for single-shot prompting, where $(x_i, \bar{y}_i) \in D_s$. Since the LLM is an autoregressive language model, we simply ask ChatGPT to provide the annotation for each query $x$ without giving $\bar{y}_i$ for zero-shot prompting. Consequently, we obtain the teacher distribution $D_t = \{(x_i, \bar{y}_i^t)\}_{i=1}^N$ and the augmented distribution $\hat{D}_t = \{(x_i, \hat{y}_i^t)\}_{i=1}^N$. During prompting, we set the temperature parameter to 1 to maximize output diversity. The reason for acquiring two distributions—one with and one without the student model's annotations—is to ensure output diversity and prevent performance collapse when the LLMs exhibit limited competence in the task. Additionally, providing step-by-step explanations has been shown to enhance LLM performance (Wei et al., 2022).

### 3.2 CONSISTENT INCONSISTENT AND INCONSISTENT SAMPLE IDENTIFICATION AND RATIO

#### 3.2.1 CONSISTENT AND INCONSISTENT (CAI) IDENTIFICATION FOR UNSUPERVISED DATASETS

After we have acquired $D_s = \{(x_i, \bar{y}_i)\}_{i=1}^N$, $D_t = \{(x_i, \bar{y}_i^t)\}_{i=1}^N$, and $\hat{D}_t = \{(x_i, \hat{y}_i^t)\}_{i=1}^N$, the first challenge still remains unresolved: assessing the annotations generated by LLMs or assigned by the student model due to the unavailability of ground truth annotations. To address this problem, we propose the **Consistent and Inconsistent Sample (CAI)** *Identification* and *Ratio*. The CAI identification aims to identify the consistent and inconsistent samples among $D_s$, $D_t$, and $\hat{D}_t$, specifically focusing on samples with consistent annotations across the student and teacher distributions. More precisely, the CAI identification utilizes annotations from the teacher model LLMs $\mathcal{T}$ and the pre-trained sentence embedder as a student model $\mathcal{S}$. Samples with the same predictions from both the student and teacher models are defined as consistent samples; otherwise, they are inconsistent samples. For each $x \in \mathcal{D}_u$, the annotation assignment process is represented by the function $h$, which assigns an output label for the student model. Specifically, the label assigned by the student model is given by $\bar{y}_{\mathcal{S}} = h(x)$ where, for a sample $x_i$, the function assigns the label of the cluster with the highest average cosine similarity: $\bar{y}_{\mathcal{S}} = h(x_i) = \bar{y}_{j*}$. For each $x \in \mathcal{D}_u$, the teacher model $\mathcal{T}$ generates a annotation: $\bar{y}_{\mathcal{T}} = \mathcal{T}(x; t)$, and $\hat{y}_{\mathcal{T}} = \mathcal{T}(x, \bar{y}; t)$, where $t$ denotes the temperature parameter controlling diversity. **Consistency Check:** If $\bar{y}_{\mathcal{S}} = \bar{y}_{\mathcal{T}} = \hat{y}_{\mathcal{T}}$, then $x \in \mathcal{C}$ (consistent samples). If $\bar{y}_{\mathcal{S}} \neq \bar{y}_{\mathcal{T}} \neq \hat{y}_{\mathcal{T}}$, then $x \in \mathcal{I}$ (inconsistent samples). We have also provided a pseudo-algorithm table as follows:

---

**Algorithm 1:** Consistent and Inconsistent Sample Identification

---

**Input:** Dataset $\mathcal{D}_u = \{x_1, x_2, \ldots, x_n\}$, Teacher Model $\mathcal{T}$, Student Model for annotation assignment $h$
**Output:** Consistent Samples $\mathcal{C}$, Inconsistent Samples $\mathcal{I}$
`Initialize`($\mathcal{C} \leftarrow \emptyset, \mathcal{I} \leftarrow \emptyset$);
**for** *each $x_i \in \mathcal{D}$* **do**
    $\mathcal{T}(x_i) \rightarrow \bar{y}_{\mathcal{T}}, \mathcal{T}(x_i, \bar{y}_i) \rightarrow \hat{y}_{\mathcal{T}}, h(x_i) \rightarrow \bar{y}_{\mathcal{S}}$;
    **if** $\bar{y}_{\mathcal{T}} == \bar{y}_{\mathcal{S}} == \hat{y}_{\mathcal{T}}$ **then**
        $\mathcal{C} \leftarrow \mathcal{C} \cup \{x_i\}$            // Consistent
    **else**
        $\mathcal{I} \leftarrow \mathcal{I} \cup \{x_i\}$            // Inconsistent

**return** $\mathcal{C}, \mathcal{I}$;

---

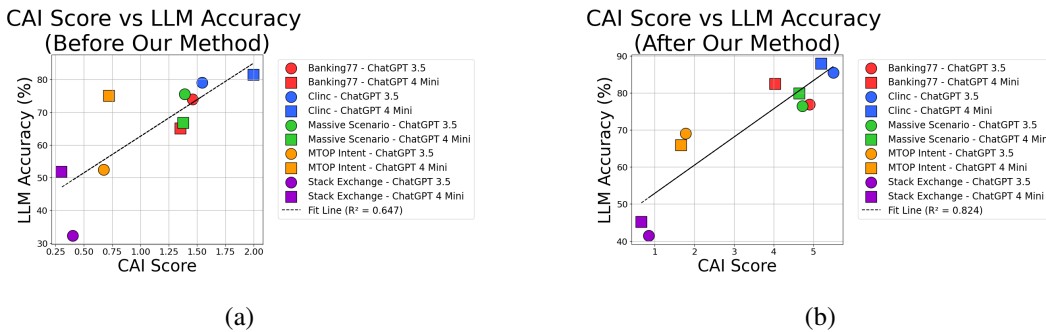

Figure 2: The above analysis shows the correlation between LLM annotation accuracy and the Consistent and Inconsistent (CAI) ratio. We also conducted statistical tests to assess the significance of this correlation. We collected the CAI ratios for (LLMs 3.5 Turbo and Student Model) and (LLMs 4.0 Mini and Student Model) across the datasets CLINC, Massive Scenario, MTOP Intent, Stack Exchange, and Banking77. Using these data, we calculated the Pearson correlation coefficients between the LLM annotation accuracies and CAI ratios and computed the associated P-values to determine the statistical significance of the observed correlations.

Nonetheless, the identification of consistent and inconsistent samples still cannot tell us about the quality of annotations generated and assigned by the LLMs and the student model.

### 3.2.2 CONSISTENT AND INCONSISTENT (CAI) RATIO FOR UNSUPERVISED DATASETS

Given the identification of consistent and inconsistent samples through our CAI identification process, we propose the Consistent and Inconsistent (CAI) ratio to evaluate LLM-generated annotations on unsupervised data with user preferences. Additionally, the CAI ratio measures the confidence of the LLM-generated outputs—in this case, the annotations. We define the size of the consistent sample set as $N_C$ and the size of the inconsistent sample set as $N_{IC}$. The CAI ratio is defined as follows:

$$\text{CAI Ratio} = \frac{N_C}{N_{IC}} \qquad (3)$$

From our observations: When the CAI Ratio $> 1$ (i.e., $N_C > N_{IC}$), the LLM-generated annotations are more certain and consistent, demonstrating higher confidence in the model's predictions. Conversely, when the CAI Ratio $< 1$ (i.e., $N_C < N_{IC}$), it reflects less certainty and consistency, suggesting the need to adjust the prompting approach or switch to a different student model for the given unsupervised dataset. Furthermore, if the CAI ratio is too low, indicating that $N_{IC}$ greatly outnumbers $N_C$, prior knowledge or additional human annotations are necessary to improve annotation accuracy. Overall, a CAI Ratio $> 1$ indicates that the LLM is more confident in its predictions, whereas a CAI Ratio $< 1$ shows that the model is less confident in its predictions.

### 3.2.3 LAW OF CONSISTENCY

We have defined the phenomenon of the higher CAI ratio showing higher LLM annotation accuracy as the Law of Consistency, stating that if both the LLM model and student model are optimal hypotheses which are $T^*$ and $S^*$ for the given dataset $D_u$, the number of identified consistent samples should outnumber the identified inconsistent samples as number of sample reach to a infinite large. We have conducted significance testing to justify our findings that the CAI ratio can serve as an indicator of LLM performance under unsupervised data with user preferences. Additionally, **Figures 2(a)** and **2(b)** demonstrate a strong positive correlation between a higher CAI ratio and higher LLM annotation accuracy, with $R^2 = 0.647$ and $R^2 = 0.824$, respectively.

### 3.3 RETROSPECTIVE LEARNING (RETROL) FOR SELF-CORRECTION OF INCONSISTENT SAMPLES (DCSC)

RetroL consists of two key components: Divide-and-Conquer Self-correction and Majority Voting via the Top-Nearest Embedding Scheme. These approaches work collaboratively to achieve self-correction of the inconsistent samples.

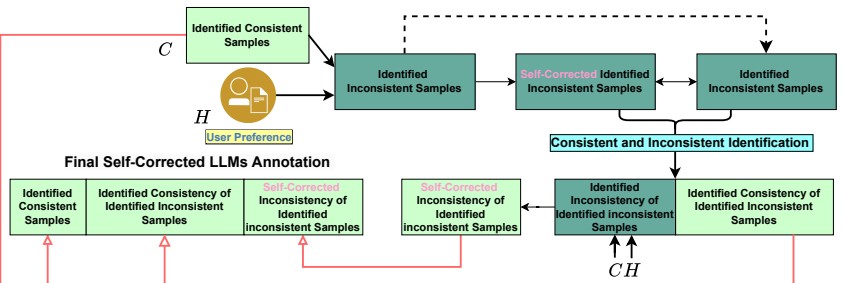

Figure 3: **Divide-and-conquer self-correction for the Inconsistent samples.** Using CAI identification, we first identify the consistent and inconsistent samples, denoted $D_u = I \cup C$. Subsequently, we apply the DCSC process to refine further inconsistency of the identified inconsistent sample where $II \in I$.(**See Section 3.3**).

### 3.3.1 DIVIDE-AND-CONQUER SELF-CORRECTION

Given the identified consistent samples ($C$), inconsistent samples ($I$) determined through CAI identification, and user preference samples, the next challenge we address is resolving the self-correction of identified inconsistent samples from $D_s$, $D_t$, and $\hat{D}_t$. Our proposed Divide-and-Conquer Self-Correction (DCSC) approach effectively addresses this challenge. We begin by leveraging the consistent samples ($C$) and the inconsistent samples ($I$), further dividing $I$ into two categories using CAI identification: $CI$ (consistency of identified inconsistent samples) and $II$ (inconsistency of identified inconsistent samples). The DCSC process consists of two rounds of self-correction (see Figure 2). In the first round of identification and self-correction, we aim to refine the identified inconsistent samples ($I$) using the consistent samples ($C$) and the user preference samples ($H$). This process results in self-corrected inconsistent samples through majority voting based on the top-nearest embedding scheme (MV-VTES). Once this correction is completed, we reapply CAI identification to the self-corrected inconsistent samples and remaining inconsistent samples. This second step identifies $II$ samples for further self-correction, incorporating user preference samples ($H$) and consistent samples ($C$). This second round completes our Divide-and-Conquer Self-Correction (DCSC) paradigm.

### 3.3.2 MAJORITY VOTING VIA TOP-NEAREST EMBEDDING SCHEME (MV-VTES)

The self-correction of inconsistent samples and inconsistency of inconsistent samples is realised by applying an MV-VTES, which includes selecting the most semantic similar example from the identified consistent samples and user-preference samples for each inconsistent sample. Choose example $(a_{top}, l_{top})$ as an positive example from $D_{(A,L)_e}$, based on highest cosine similarity score with $x$ to be fed into $G_t(x, a_{top}, l_{top})$. Given an query which is denoted as $x$, our goal is to find the positive example $\{a_{\text{top}}, l_{\text{top}}\}$ in $D_{(A,L)_e}$ that has the highest cosine similarity score with $x$.

$$\{(a_i, \bar{y}_i)\}_{i=1}^K = \underset{(a_i, l_i) \in D_{(A,L)_e}}{\arg \text{top-}K} \left( \frac{\mathcal{S}(a_i) \cdot \mathcal{S}(x)}{\|\mathcal{S}(a_i)\| \|\mathcal{S}(x)\|} \right) \tag{4}$$

given the selected top-k positive samples, the final annotation is assigned with majority voting and averaging. Basically, the annotations in $\{a_i, \bar{y}_i\}$, which occurs the most frequently, will be voted as the final prediction. Let $\bar{y}_i$ denote the annotation associated with $a_i$ in the top-K samples. For each $x \in I$, there is a corresponding $\{(a_i, \bar{y}_i)\}_{i=1}^K$ applying our self-reflection search algorithm. For the top-k selected samples for $x_i$, there will be a set of possible annotations $A = \{a_1, a_2, \ldots, a_k\}$ corresponding to each top-k selected sample. For the given set $A$, we compute the frequency $n_a$, denote as $n_a = \sum_{i=1}^K \mathbf{1}_{\{\bar{y}_i = a\}}$, of each annotation $a \in A$ and assign the annotation of $A$ with the highest frequency as the final annotation $\hat{y}$ for $x$. The formula is defined as follows:

$$\hat{y} = \underset{a \in A}{\arg \max} \; n_a \tag{5}$$

The cosine similarity score measures the semantic embedding similarity between the Query $x_i$ and Sample $a$. The selected Sample with the highest cosine similarity score is considered a positive

sample. Unlike traditional data-centric methods, our philosophy is to use the extracted information to correct inconsistent samples. Traditional approaches, such as data pruning (Yang et al., 2022; Liu et al., 2020) or noisy-teacher and student distillation (Chen et al., 2024), focus on extracting only the most correct or informative data. We believe all representative samples (consistency and inconsistency) should be considered to train a more robust and generalizable model. Frequently, the inconsistent samples hinder model performance and annotation accuracy the most. Higher-quality annotation and improved model performance cannot be acquired without correcting incorrect annotations among these inconsistent samples.

## 4 EXPERIMENTS

### 4.0.1 BASELINES

**Using Only Student:** We utilise a student model using user preference samples to assign initial annotation for unsupervised data with our proposed preference-based annotation scheme. Applying a pre-trained student model to annotate unsupervised data according to user preferences is a cost-effective approach compared to crowd-sourcing or even LLMs.

**Using Only LLMs:** As our second baseline, we use LLMs *(ChatGPT 3.5 and ChatGPT 4o mini)* in a zero-shot setting. The categories defined by user preferences are provided during prompting. The application of LLMs for unsupervised textual data is considered affordable, but it might be unreliable if the generated outputs are incorrect or inconsistent.

**Student (Our) and LLMs (ChatGPT 4o Mini and ChatGPT 3.5):** We use the Student model with our proposed **hint-based majority voting approach** to assign annotation for each unsupervised data. Then, we use it as a demonstration to help LLMs to generate annotation.

**A Consistent Sample of Student and Teacher Knowledge Distillation:** A special case of our methodology is the distillation of student-teacher knowledge using consistent samples. In order to enhance downstream model generalisation, we systematically identify inconsistent samples and exclude them from the training process of pretrained BERT as the student model. This is comparable to our retrospective learning approach; rather than reconfiguring the student model, we implement self-correction and reassign annotations to the inconsistent samples.

**Clustering Approach:** Our proposed Majority Voting via the Top-Nearest Embedding Scheme is a new clustering method for unsupervised data annotation. Therefore, we include Zhang et al. (2023) as one of our baselines, which is the state-of-the-art (SOTA) in current clustering methods, for comparison.

**Self-Refine & Reflexion Prompting Methods:** We have added two methods as baselines: Self-Refine and Reflexion. Self-refine is designed to improve initial output through iterative rounds of self-correction (Madaan et al., 2024). Reflexion aims to achieve self-correction through LLMs' own evaluations and incorporates feedback from internal or external tools (Shinn et al., 2024). Both techniques largely depend on the LLMs' ability to effectively generate accurate annotations. (Further details can be found in Appendix A.4)

### 4.1 EVALUATION METRICS

The evaluation of our method and baselines is based on two metrics: *Annotation Accuracy* and the CAI ratio evaluation (change from the *initial CAI ratio* to the *after CAI ratio*). The first metric assesses the accuracy to evaluate the effectiveness of our method. The second metric evaluates whether the size of inconsistent samples has decreased and consistent samples have increased after applying retrospective learning.

### 4.2 DATASET

In this paper, we evaluate a wide range of open-source textual datasets. These include Bank77, CLINC (Intent), MTOP (Intent), Massive (Intent), StackExchange and Reddit (Topic) (Geigle et al.,

| Task | Name | #clusters | #data(small) | #data(large) |
|---|---|---|---|---|
| Intent | Bank77 | 77 | 3,080 | 10,003 |
| | CLINC(I) | 150 | 4,500 | 15,000 |
| | MTOP(I) | 102 | 4,386 | 15,638 |
| | Massive(I) | 59 | 2,974 | 11,510 |
| Type | FewRel | 64 | 4,480 | 40,320 |
| | StackEx | 121 | 4,156 | 50,000 |
| Topic | Reddit | 50 | 3,217 | 50,000 |
| Domain | Massive Scenario | 18 | 2,974 | 11,514 |

Table 1: Dataset Summary

2021), and Few Rel Nat (Type). We also utilize the Massive Intent dataset with some modifications, following the approach in (Zhang et al., 2023). Since we are working in an unsupervised textual data setting, we directly use the small-scale version of each dataset for testing. Intent discovery (Zhang et al., 2021; 2022) explores unknown intents in unsupervised utterance datasets. Bank77 (Casanueva et al., 2020) is a banking dataset that focuses on fine-grained intent classification within a single domain. CLINC (I), Massive (I), and MTOP (I) are intent-based datasets where "I" refers to intent (Larson et al., 2019; FitzGerald et al., 2022; Li et al., 2020). Each dataset is available in small-scale and large-scale versions; we use i.i.d. user preference samples from the large-scale versions.

## 4.3 EXPERIMENTAL RESULT

### 4.3.1 EXPERIMENTAL ANALYSIS

**Chatgpt 3.5-Turbo:** Based on the experimental results from Table 2 and Table 3, we have two key findings. First, our method (RetroL) outperformed on three datasets: CLINC (**+4.17%**), Massive Scenario (**+0.88%**), and Bank77 (**+2.99%**). For the MTOP Intent and StackExchange datasets, our method (RetroL) outperformed the Only Student baseline by **+16%** on MTOP and **+9.18%** on StackExchange. It also improved over the Only LLMs (ChatGPT 3.5) baseline by **+4.11%** on MTOP and **+11.35%** on StackExchange, showing that our method can achieve greater improvements over each model. Additionally, the student-teacher knowledge distillation (KD) with consistent samples achieved the highest annotation accuracy on the MTOP Intent dataset. **Chatgpt 4o-mini:** Based on the experimental results from table 2 and table 3, there are two aspects of findings first is that RetroL (Our) has outperformed all baseline methods on three datasets, which are Clinc(**+2.7%**), Massive Scenario (**+0.58%**), and Bank77(**+7.06%**). The student-teacher knowledge distillation (KD) with consistent samples achieved the highest annotation accuracy on the MTOP Intent dataset. In practice, Retrospective Learning and student-teacher knowledge distillation with consistent samples can be used interchangeably for improved annotation accuracy. On the Llama 8B Instruct model Touvron et al. (2023), our proposed RetroL has outperformed all baselines, demonstrating a significant improvement in accuracy and CAI scores. Notably, despite the relatively poor accuracy of the Llama 8B model, our method shows remarkable robustness by consistently outperforming both the Llama 8B model and student models. This highlights the adaptability and reliability of our approach.

## 4.4 CAI RATIO EVALUATION

The following table (Figure 4) shows the changes in the number of consistent and inconsistent samples identified before and after applying our proposed retrospective learning approach. For Banking77 (**1.45** ⇒ **4.99**), CLINC (**1.44** ⇒ **5.74**), Massive_Scenario (**1.38** ⇒ **4.88**), MTOP_INTENT (**0.67** ⇒ **1.65**), and StackExchange (**0.40** ⇒ **0.86**), highlighting the significant improvement in the CAI ratio resulting from improvement of corresponding annotation accuracy across datasets. Table 5 in the appendix illustrates the improvements in both the CAI ratio and accuracy after applying our proposed retrospective learning approach. For Banking77, the CAI ratio improved from **1.35** to **4.03**, with an accuracy increase from **65.12%** to **82.45%**. Similarly, for CLINC, the CAI ratio rose from **1.99** to **5.20**, and accuracy improved from **81.44%** to **87.93%**. For the Massive Scenario dataset, the CAI ratio increased from **1.38** to **4.65**, with accuracy going from **66.83%** to **80.18%**. However, for the MTOP Intent dataset, although the CAI ratio increased from **0.72** to **1.66**, the annotation accuracy of LLMs performed worse than Only LLMs (ChatGPT 4o Mini), dropping from **75.03%** to **67.10%**. Similarly, for StackExchange, while the CAI ratio increased from **0.30** to **0.66**, the accuracy of Only LLMs (**51.90%**) outperformed the annotation accuracy of our RetroL approach, which was **45.22%**. The explanation for this is that, by examining the other datasets, we can observe that only when the

| Datasets | Only Student Model (Our) | Only LLMs (ChatGPT 3.5) | Student (Our) & LLM (Chat-GPT 3.5) | Clustering Based Method (Zhang et al., 2023) | Student & Teacher KD (Our) | Retrospective Learning (Our)(%) | CAI Ratio & (Before & After) |
|---|---|---|---|---|---|---|---|
| **Clinc** | 79.01 | 66.58 | 76.82 | 78.58 | 81.32 | **85.49** | 1.55 |
| Std Dev | ±1.08 | ±3.36 | ±1.51 | ±0.41 | ±0.46 | ±0.19 | 5.50 |
| **Massive_Scenario** | 75.55 | 60.89 | 70.23 | 60.85 | 69.25 | **76.43** | 1.39 |
| Std Dev | ±1.76 | ±0.62 | ±1.64 | ±4.33 | ±0.03 | ±2.47 | 4.72 |
| **Mtop Intent** | 52.49 | 64.95 | 55.12 | 37.22 | **79.57** | 69.06 | 0.68 |
| Std Dev | ±2.52 | ±0.21 | ±3.08 | ±1.18 | ±0.42 | ±1.10 | 1.78 |
| **StackExchange** | 32.27 | 30.10 | 30.92 | **47.75** | 29.76 | 41.45 | 0.40 |
| Std Dev | ±0.65 | ±0.10 | ±2.21 | ±1.24 | ±0.19 | ±2.56 | 0.85 |
| **Banking77** | 73.93 | 60.29 | 73.15 | 71.20 | 70.11 | **76.92** | 1.46 |
| Std Dev | ±0.81 | ±1.33 | ±1.70 | ±1.59 | ±0.12 | ±0.02 | 4.91 |
| **Reddit** | 51.73 | 51.12 | 51.64 | 57.02 | 43.90 | **58.77** | 0.50 |
| Std Dev | ±0.62 | ±1.27 | ±0.18 | ±1.59 | ±1.59 | ±0.29 | 1.40 |
| **Few Rel Nat** | 35.35 | 32.87 | 37.37 | **51.22** | 49.24 | 44.88 | 0.28 |
| Std Dev | ±0.016 | ±1.72 | ±0.13 | ±1.43 | ±0.63 | ±0.05 | 0.89 |
| **Massive_Intent** | 61.80 | 71.52 | 64.54 | 60.69 | 73.41 | **71.72** | 1.62 |
| Std Dev | ±1.04 | ±0.95 | ±0.024 | ±0.024 | ±1.843 | ±0.40 | 2.81 |

Table 2: **Chatgpt-3.5 Turbo (Closed-source LLMs):** "Before Correction" means before applying our Retrospective Learning. The highest accuracy for each dataset is highlighted

| Datasets | Only Student Model (Our) | Only LLMs (Chatgpt-4o mini) | Student (Our) &LLM (Chatgpt-4o mini) | Clustering Based Method (Zhang et al., 2023) | Student & Teacher KD (Our) | Retrospective Learning (Our)(%) | CAI Ratio & (Before & After) |
|---|---|---|---|---|---|---|---|
| **Clinc** | 79.01 | 81.44 | 78.58 | 78.58 | 85.23 | **87.93** | 2.06 |
| Std Dev | ± 1.08 | ± 0.44 | ± 1.35 | ± 0.41 | ±0.98 | ± 0.53 | 5.20 |
| **Massive_Scenario** | 75.55 | 66.83 | 77.62 | 60.85 | 79.60 | **80.18** | 1.37 |
| Std Dev | ±1.76 | ± 1.31 | ± 0.74 | ± 4.33 | ±0.85 | ± 0.45 | 4.65 |
| **Mtop Intent** | 52.49 | 75.03 | 57.01 | 37.22 | **80.16** | 67.10 | 0.74 |
| Std Dev | ± 2.52 | ± 1.35 | ± 0.37 | ± 1.18 | ± 0.85 | ± 0.32 | 1.66 |
| **StackExchange** | 32.27 | **51.90** | 45.49 | 47.75 | 35.63 | 45.22 | 0.31 |
| Std Dev | ± 0.65 | ± 0.75 | ± 0.94 | ± 1.24 | ± 0.51 | ± 0.15 | 0.66 |
| **Banking77** | 73.93 | 65.12 | 75.39 | 71.20 | 73.56 | **82.45** | 1.36 |
| Std Dev | ± 1.56 | ± 0.30 | ± 0.32 | ± 1.59 | ± 0.20 | ± 0.48 | 4.03 |
| **Reddit** | 51.73 | 53.25 | 57.40 | 57.02 | 44.47 | **60.94** | 0.51 |
| Std Dev | ± 0.62 | ± 0.35 | ± 1.96 | ± 1.59 | ± 0.69 | ± 0.11 | 1.90 |
| **Few Rel Nat** | 35.35 | 37.11 | 38.87 | **51.22** | 49.53 | 44.94 | 0.26 |
| Std Dev | ± 0.016 | ± 0.03 | ± 1.88 | ± 1.43 | ± 0.35 | ± 0.02 | 0.9 |
| **Massive_Intent** | 61.80 | 66.02 | 76.93 | 60.69 | **78.93** | 72.49 | 1.47 |
| Std Dev | ± 1.04 | ± 0.35 | ± 1.05 | ± 0.024 | ± 0.50 | ± 0.40 | 3.3 |

Table 3: **Chatgpt-4o mini (Closed-source LLMs):** "Before Correction" means before applying our Retrospective Learning. The highest accuracy for each dataset is highlighted.

| Datasets | Only Student Model (Our) | Only LLMs (Llama-8B-Instruct) | Student (Our) & LLM (Llama-8B-Instruct) | Student & Teacher KD (Our) | Retrospective Learning (Our)(%) | CAI Ratio & (Before & After) |
|---|---|---|---|---|---|---|
| **Clinc** | 79.01 ±1.08 | 32.49 ±6.73 | 69.40 ±7.28 | 63.41 ±3.19 | **82.43**±0.20 | 0.56⇒4.43 |
| **Massive_Scenario** | 75.55 ±1.76 | 43.52 ±1.85 | 66.74 ±0.98 | 70.06±1.12 | **78.13** ±0.74 | 0.67⇒4.88 |
| **Mtop Intent** | 52.49 ±2.52 | 34.17 ±6.70 | 48.23 ±0.25 | **66.39** ±0.70 | 63.39 ±1.47 | 0.35⇒1.46 |
| **StackExchange** | 32.27 ±0.65 | 11.02 ±2.78 | 26.26 ±2.16 | 16.03 ±0.13 | **38.88** ±0.27 | 0.23⇒0.53 |
| **Banking77** | 73.93 ±1.56 | 33.06 ±1.92 | 69.66 ±1.74 | 64.29 ±1.24 | **77.71** ±0.25 | 0.68⇒4.20 |
| **Reddit** | 51.73 ±0.62 | 36.31 ±0.97 | 46.00 ±2.51 | 40.29±0.55 | **58.81** ±0.28 | 0.33⇒1.58 |
| **Few Rel Nat** | 35.35 ±0.016 | 14.25 ±0.36 | 30.07 ±4.45 | 31.80±0.34 | **42.92** ±0.06 | 0.13⇒0.85 |
| **Massive_Intent** | 61.80 ±1.04 | 45.41 ±0.06 | 56.03 ±0.08 | 67.49 ±0.10 | **67.75** ±0.43 | 0.73⇒2.87 |

Table 4: **Meta-Llama 3-8B Instruct (Open-Source Light-Weight LLMs):** "Before Correction" means before applying our Retrospective Learning. The highest accuracy for each dataset is highlighted

CAI ratio increases by a large margin can we confidently claim there is a significant improvement. However, if the CAI ratio improvement is very small, as in the case of StackExchange (an increase of only 0.36), this suggests that either the student model chosen or the prompt given to the LLMs was not optimal. Therefore, while the CAI ratio remains a good indicator, the improvement from the CAI ratio must be sufficiently large to reliably indicate improved performance.

### 4.4.1 STATISTICAL TEST FOR THE CORRELATION BETWEEN CAI SCORES AND AVERAGE LLM ACCURACY

We performed a Pearson correlation analysis to examine the relationship between CAI ratios and average LLM accuracy after implementing our suggested approach. Our objective was to investigate the possibility of a relationship between higher LLM annotation accuracy and larger CAI ratios. The *null hypothesis* in **test** asserts that there is no meaningful positive correlation between CAI scores

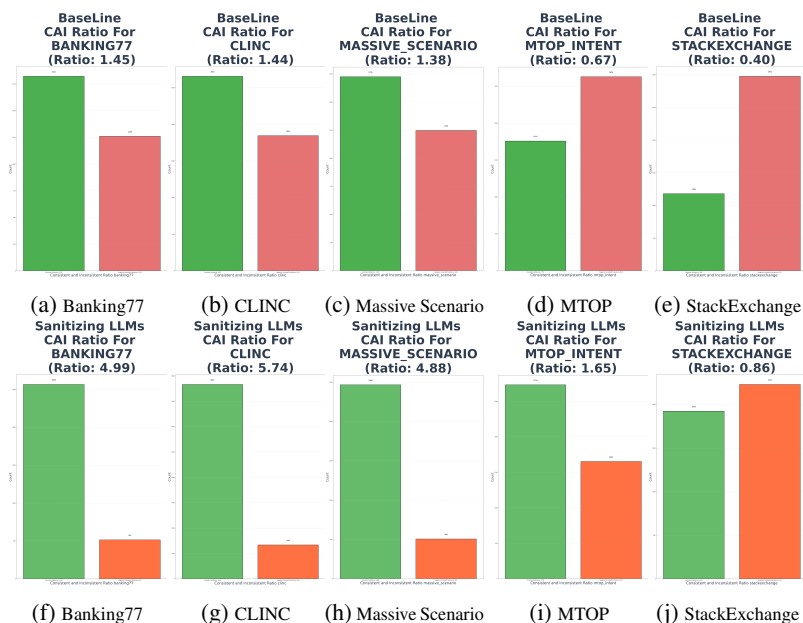

(a) Banking77    (b) CLINC    (c) Massive Scenario    (d) MTOP    (e) StackExchange

(f) Banking77    (g) CLINC    (h) Massive Scenario    (i) MTOP    (j) StackExchange

Figure 4: Performance comparison based on LLMs (Chatgpt 3.5) and student model (MINILM Wang et al. (2020)) across 5 different datasets. The **first row** presents the CAI ratio before applying our sanitizing LLMs for the Student+LLMs (Chatgpt 3.5) baseline, while the **second row** shows the results after applying our proposed sanitizing LLMs, demonstrating a significant reduction in the number of inconsistent samples. (More Details are in the appendix A2 and A8.)

and LLMs annotation accuracy, while the *alternative hypothesis* postulates a substantial positive correlation. The following tables display the statistics. There was a strong positive correlation between the CAI ratio and the LLM accuracy, with a p-value of about 0.005 and 0.00035. The value of r was about 0.805, and the value of r was about 0.903. This implies that the correlation is statistically significant, further suggesting that higher CAI ratios are associated with higher Average LLM Accuracy. Full details of the analysis are provided in Appendix A.2.

| Metric | Pearson Correlation | p-value |
|---|---|---|
| CAI Ratio vs. Annotation Accuracy (Before RetroL (Our)) | 0.805 | 0.005 |
| CAI Ratio vs. Annotation Accuracy (After RetroL (Our)) | 0.903 | 0.00035 |

Table 5: Pearson Correlation Between CAI Ratio and Annotation Accuracy (Before and After RetroL (Our)

## 5 DISCUSSION

This paper proposes a retrospective learning framework to address two critical challenges in unsupervised data tasks involving user preferences: evaluation and self-correction. To tackle the evaluation challenge, we introduce consistent and inconsistent (CAI) identification along with the consistent and inconsistent (CAI) ratio, an effective evaluation metric for unsupervised data with user preferences. Building on this CAI identification and ratio, we propose a Divide-and-Conquer Self-Correction paradigm that leverages consistent samples to iteratively self-correct identified inconsistent samples. Our approach addresses the self-correction challenge by achieving higher annotation quality compared to teacher and student models, without relying on external knowledge.

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

## A APPENDIX / SUPPLEMENTAL MATERIAL

### A.1 CLUSTERING OPERATIONS IN RETROSPECTIVE LEARNING

---

**Algorithm 2:** Clustering Operations in Retrospective Learning

---

**Input:** Pre-trained student model $S$, Annotated set $H = x_1, x_2, \ldots, x_s$, Unlabelled dataset $D_u$, Number of clusters $k$

**Output:** Student-annotated dataset $D_s$

Extract Embeddings

**for** *each $x_i \in H$* **do**
    Compute embedding: $e_i = S(x_i)$;

Cluster User-Preference Samples

Partition $H$ into $k$ clusters $C_1, C_2, \ldots, C_k$ using label set $\mathcal{Y}$ such that: Assign Annotations to Unlabelled Data

**for** *each $x_i \in D_u$* **do**
    **for** *each $x_i \in D_u$* **do**
        Compute embedding: $e_i = S(x_i)$; **for** *each cluster $C_j$* **do**
            Compute Average Similarity (AS):
        Assign label of cluster $C_{j^*}$ with highest AS to $x_i$: $\bar{y}_i = \bar{y}_{j^*}$;

Construct Annotated Dataset **return** $D_s$;

---

The clustering operation is performed using the semantic similarity score (6) and majority voting based on the top-nearest embedding scheme (4). This process plays a critical role in our retrospective learning framework, as it aligns annotations with user-defined preferences. By assigning annotations in this manner, the CAI (Consistent Annotation Identification) method is employed to self-correct identified inconsistent samples through the divide-and-conquer self-correction (DCSC) process, an iterative self-correction mechanism. Our approach supports user-defined preferences and is applicable to a wide range of large NLP datasets. It achieves this by self-correcting annotations in identified inconsistent samples, a process that is particularly crucial for unsupervised learning tasks incorporating user preferences.

#### A.1.1 EXTRACTING EMBEDDING FROM STUDENT MODEL

The semantic features of text inputs are obtained by exploiting a pre trained student model (MiniLM) for generating dense vector representations. Given a sample $x_i$ which is fed into an embedding function of the student model $S(x_i)$ to compute an embedding $e_i$.

#### A.1.2 USER-PREFERENCE SAMPLE CLUSTERING

A small size of annotated set of user-preference samples $H = \{x_1, ..., x_s\}$ is provided. It is partitioned into $k$ clusters, and each cluster is defined as $C_1, C_2, ..., C_k$ using a predefined label $y_j \mathcal{Y}$ reflecting the preference of user annotation. Moreover, each cluster does not overlap with other clusters such that $(C_i \cap C_j = \emptyset, \forall i \neq j)$, and $\bigcup_{i=1}^{k} C_i = H$.

#### A.1.3 ANNOTATION ASSIGNMENT FOR UNLABELLED DATA USING STUDENT MODEL

For every unlabelled sample $x_i \in D_u$, we will acquire feature embedding $e_i = S(x_x)$ to compute the semantic similarity to samples in each cluster $C_j$ using Average Similarity which is defined as follows:

$$AS(e_i, C_j) = \frac{1}{k} \sum_{e \in \text{Top-}k(C_j, e_i)} \frac{e_i \cdot e}{\|e_i\| \|e\|}, \tag{6}$$

where $e_i$ is the embedding for $x_i$ and $e$ represents the embedding of each sample in each cluster $C_j$. where Top-$k(C_j, e_i)$ represents the subset with top $k$ embedding cosine similarity score from $C_j$. Top-$k(C_j, e_i) = \{e \in C_j \mid AS(e_i, e)$ is among the top $k$ in $C_j\}$. Based on the calculated cosine similarity, examples which are most similar to the $e_i$ and calculated average cosine similarity for the top selected sample for each cluster. In our experiment, we set the $k$ to five. Lastly, for the annotation process, we assign the label of the cluster $C_j$ with the highest average cosine similarity score to

the unlabelled sample $x_i$. The cluster $C_{j^*}$, which has the highest average cosine similarity with the embedding $e_i$ of a sample $x_i$, is defined as:

$$C_{j^*} = \arg\max_{C_j} \text{AS}(e_i, C_j)$$

where $\text{AS}(e_i, C_j)$ is the average cosine similarity of $e_i$ with the embeddings in $C_j$. The annotation $\bar{y}_{j^*}$ associated with $C_{j^*}$ is then assigned to $x_i$, i.e., $\bar{y}_i = \bar{y}_{j^*}$. This process is represented by the annotation assignment function $h(x_i)$. The final student-annotated dataset is:

$$D_s = \{(x_i, \bar{y}_i)\}_{i=1}^N$$

where each $\bar{y}_i$ represents the user preference-based annotation for the corresponding sample $x_i$.

### A.1.4 ANNOTATION ASSIGNMENT FOR UNLABELLED DATA USING TEACHER MODEL (LLMs)

With the acquired dataset $D_s = \{(x_i, \bar{y}_i)\}_{i=1}^N$, we further exploit LLMs using zero-shot prompting (without including annotations from the student) and single-shot prompting (including annotations from the student) through a group prompting method to provide annotations for each $x_i$. We define the annotations as $\bar{y}_i^t = T(x_i)$ for zero-shot prompting and $\hat{y}_i^t = T(x_i, \bar{y}_i)$ for single-shot prompting, where $(x_i, \bar{y}_i) \in D_s$. Since the LLM is an autoregressive language model, we simply ask ChatGPT to provide the annotation for each query $x$ without giving $\bar{y}_i$ for zero-shot prompting. Consequently, we obtain the teacher distribution $D_t = \{(x_i, \bar{y}_i^t)\}_{i=1}^N$ and the augmented distribution $\hat{D}_t = \{(x_i, \hat{y}_i^t)\}_{i=1}^N$. During prompting, we set the temperature parameter to 1 to maximize output diversity. The reason for acquiring two distributions—one with and one without the student model's annotations—is to ensure output diversity and prevent performance collapse when the LLMs exhibit limited competence in the task. Additionally, providing step-by-step explanations has been shown to enhance LLM performance Wei et al. (2022).

### A.1.5 IDENTIFICATION OF CONSISTENT AND INCONSISTENT SAMPLES

Given an unsupervised dataset we do not have access to ground truth annotation for assessment of acquired annotation quality. Therefore, we propose the CAI ratio, a novel metric for unsupervised tasks with user preference to evaluate the performance of LLMs and student models on the given task. In addition, the annotation assigned is not perfect and consists of annotation corruption. To address the potential annotation corruption, we propose consistent and inconsistent identification methods. *We have discussed the details of the CAI ratio and CAI identification in section 3.2 of the main paper.*

### A.1.6 MAJORITY VOTING VIA TOP-NEAREST EMBEDDING SCHEME (MV-VTES)

To correct the misaligned annotation of the inconsistent samples, MV-VTES is proposed. The key idea is to exploit the identified consistent sample and user-preference sample to self-correct the incorrectly assigned annotation on the inconsistent samples. This is based on our observation that the identified consistent sample and inconsistent sample that, we observe consistent sample has much higher accuracy than the inconsistent sample. For more details, please see section 3.3, RETROSPECTIVE LEARNING (REL) FOR SELF-CORRECTION OF INCONSISTENT SAMPLES (DCSC), on the main page.

### A.1.7 DIVIDE-AND-CONQUER SELF-CORRECTION (DCSC) APPROACH

---

**Algorithm 3:** Divide-and-Conquer Self-Correction (DCSC) Approach

---

**Result:** Self-corrected inconsistent samples
**Input**: Consistent samples $C$, Inconsistent samples $I$, Embedding function $H$, User-preference samples;
**Divide Inconsistent Samples**;
Divide $I$ into two categories using CAI identification:

    1. $CI$: Consistent identified inconsistent samples

    2. $II$: Inconsistent identified inconsistent samples

**Round 1: Identification and Self-Correction**;
**for** *each sample in $CI$* **do**
    Identify the top-nearest embeddings using $H$;
    Select the most semantically similar examples from $C$;
    Apply Majority Voting via Top-Nearest Embedding Scheme (MV-VTES) to self-correct the
    sample;
**end**
**Round 2: Re-identification and Further Self-Correction**;
Apply CAI identification again to the self-corrected samples to update $CI$ and $II$;
**for** *each sample in updated $II$* **do**
    Incorporate user-preference samples and $C$;
    Identify top-nearest embeddings using $H$;
    Select the most semantically similar examples from $C$ and user-preference samples;
    Apply MV-VTES to self-correct the sample;
**end**
**Output**: Updated and self-corrected inconsistent samples;

---

## A.2 CAI SCORES AND LLMS ACCURACY

| Dataset | CAI Before | Accuracy Before (%) | CAI After | Accuracy After (%) |
|---------|-----------|---------------------|-----------|---------------------|
| Banking77 | 1.460 | 73.93 | 4.905 | 76.92 |
| Clinc | 1.545 | 79.01 | 5.500 | 85.49 |
| Massive Scenario | 1.390 | 75.55 | 4.720 | 76.43 |
| MTOP Intent | 0.675 | 52.49 | 1.775 | 69.06 |
| Stack Exchange | 0.400 | 32.27 | 0.845 | 41.45 |

Table 6: Performance Metrics for ChatGPT 3.5 Before and After Applying Our Method

| Dataset | CAI Before | Accuracy Before (%) | CAI After | Accuracy After (%) |
|---------|-----------|---------------------|-----------|---------------------|
| Banking77 | 1.350 | 65.12 | 4.030 | 82.45 |
| Clinc | 1.995 | 81.44 | 5.195 | 87.93 |
| Massive Scenario | 1.375 | 66.83 | 4.645 | 80.18 |
| MTOP Intent | 0.720 | 75.03 | 1.655 | 67.10 |
| Stack Exchange | 0.300 | 51.90 | 0.660 | 45.22 |

Table 7: Performance Metrics for ChatGPT 4o Mini Before and After Applying Our Method

### A.2.1 STATISTICAL INFERENCE

We have conducted a two-tailed hypothesis test based on the CAI ratio before, LLMs accuracy before and CAI ratio after, and LLMs accuracy after from Table 4 and Table 5. The **test** is to prove that there is a strong positive relationship between high CAI scores and high LLMs annotation accuracy. We have performed a Pearson correlation, the correlation coefficient $r$ is calculated as:

$$r = \frac{\sum_{i=1}^{n}(x_i - \bar{x})(y_i - \bar{y})}{\sqrt{\sum_{i=1}^{n}(x_i - \bar{x})^2}\sqrt{\sum_{i=1}^{n}(y_i - \bar{y})^2}}$$

Table 8: Performance Comparison of ChatGPT 3.5 and ChatGPT 4o Mini on additional Datasets

| Dataset | ChatGPT 3.5 | | ChatGPT 4o Mini | |
|---|---|---|---|---|
| | CAI (Before) | Accuracy (Before) | CAI (After) | Accuracy (After) |
| Reddit | 0.50 | 51.54 | 0.43 | 51.49 |
| Go Emotion | 0.12 | 21.94 | 0.13 | 31.84 |
| Few Rel Nat | 0.28 | 37.37 | 0.26 | 32.87 |
| Few Nerd Nat | 0.43 | 32.63 | 0.32 | 46.74 |
| Massive Intent | 1.63 | 64.54 | 1.47 | 71.52 |
| Reddit (After) | 1.8 | 60.94 | 0.74 | 60.94 |
| Go Emotion (After) | 0.32 | 25.69 | 0.31 | 23.56 |
| Few Rel Nat (After) | 0.88 | 44.56 | 0.9 | 44.94 |
| Few Nerd Nat (After) | 0.92 | 33.74 | 0.9 | 34.11 |
| Massive Intent (After) | 3.27 | 71.72 | 2.79 | 72.49 |

where $x_i$ symbolises the CAI ratios. $y_i$ denotes the LLM annotation accuracies. $\bar{x}$ and $\bar{y}$ are the average mean of $x_i$ and $y_i$, accordingly. $n$ is the number of samples we have used for evaluation. To assess the statistical significance, we use a hypothesis test for the correlation coefficient, calculating a t-statistic (Schober et al., 2018):

$$t = r\sqrt{\frac{n-2}{1-r^2}}$$

The P-value is then calculated from the t-distribution with $n-2$ degrees of freedom.

| Metric | Pearson Correlation | p-value |
|---|---|---|
| Before | 0.805 | 0.005 |
| After | 0.903 | 0.00035 |

Table 9: Pearson Correlation Results for CAI and Accuracy (Before and After)

Table 10: Pearson Correlation Results for CAI and Accuracy for additional datasets on ChatGPT 3.5 Turbo and ChatGPT 4o mini

| Metric | Pearson Correlation | P-value |
|---|---|---|
| Before | 0.8742509926234142 | 0.0009373655969838773 |
| After | 0.8520502618079272 | 0.0017465070325696618 |

Table 11: Pearson Correlation Results for CAI and Accuracy (Before and After) on Meta-Llama-3-8B-instruct

| Metric | Pearson Correlation | P-value |
|---|---|---|
| Before | 0.8118066946938405 | 0.014399601133794526 |
| After | 0.9180808900843687 | 0.001291289343334756 |

### A.2.2 BEFORE APPLYING THE METHOD:

The Pearson correlation coefficient is 0.805, indicating a strong positive linear relationship between CAI and Accuracy. The p-value is 0.005, which is statistically significant (below the typical threshold of 0.05). This implies that the positive correlation between the CAI ratio and Accuracy before and after applying our method is not a random event, and higher CAI scores are associated with higher Accuracy. The p-value is 0.0009373655969838773, which is statistically significant (below the typical threshold of 0.05) for the additional datasets. The p-value is 0.014399601133794526, which

is statistically significant (below the typical threshold of 0.05) for Meta-Llama-3-8B-instruct on all datasets.

### A.2.3   AFTER APPLYING THE METHOD:

The Pearson correlation coefficient is 0.903, showing an even stronger positive correlation between CAI and Accuracy after applying the method. A larger CAI ratio and higher annotation produced by LLMs are extremely statistically significant, according to the p-value of 0.00035. This implies that the relationship between CAI and Accuracy is even more evident after using the approach, showing a more linear relationship where increases in CAI are more directly correlated with increases in Accuracy. In both stages (Before and After applying the method), the results display statistically significant correlations ($p < 0.05$), showing strong positive relationships between CAI scores and LLM accuracy. Tables 9, 10, and 11 show that all the P-values of the Pearson correlation are statistically significant.

| Metric | Before | After | Source |
|---|---|---|---|
| **Slope** | 22.337 | 7.603 | Regression |
| **Intercept** | 40.317 | 45.280 | Regression |
| **R-value** | 0.805 | 0.908 | Pearson Correlation |
| **R-squared (from Pearson)** | 0.647 | 0.824 | Pearson Correlation |
| **R-squared (from Regression)** | 0.647 | 0.824 | Regression |
| **P-value** | 0.00501 | 0.00029 | Regression |
| **Standard Error** | 5.830 | 1.244 | Regression |

Table 12: Comparison of R-squared Values from Pearson Correlation and Linear Regression Before and After Applying the Method. Note: $R^2$ emphasis on variation of LLMs annotation accuracy is explained by CAI ratio. The Pearson correlation shows the strength of the linear correlation. Therefore, on the main page, we have shown the $R^2$.

### A.2.4   WHY TEACHER-STUDENT COLLABORATION IS ESSENTIAL FOR RETROSPECTIVE LEARNING

In unsupervised learning tasks that rely on user preferences to align data annotations with expectations—where the competency of the teacher model (LLMs) is uncertain and no external knowledge is available, the key challenge lies in evaluating the LLMs generated annotation and enabling mechanisms for self-correction. To address this, we propose a novel approach termed retrospective learning, a self-supervised framework designed to facilitate self-correction and self-assessment of annotations generated by large language models (LLMs).

Our methodology leverages a student model to collaborate with a teacher model of uncertain competency. By introducing the consistent and inconsistent (CAI) ratio , we quantify and identify consistent and inconsistent samples, thereby enhancing the performance of both the student and teacher models through iterative refinement.

To further demonstrate the efficacy of our approach, we conducted experiments utilizing the Meta-8B Instruct lightweight LLM as a low-competency noisy teacher. This setup, in conjunction with the student model, underscores the robustness of our framework and highlights the pivotal role of the student model in effectively managing scenarios involving noisy teachers. Given an unsupervised learning task we do not know how competent the teacher deployed to the particular learning task, and there is no external knowledge. How can we evaluate and enable the self-correction for the unsupervised dataset? To tackle this problem, we proposed retrospective learning, which is a self-supervised strategy which allows us to self-correct and self-evaluate the LLMs generated annotation. To achieve this, we introduce the student model to collaborate with an unknown competency of a teacher model to acquire a CAI ratio and identify the consistent and inconsistent samples to improve the performance of students and teachers.

### A.2.5   THE ROLE OF STUDENT MODEL IN RETROSPECTIVE LEARNING

The inclusion of the student model is essential as it provides a safeguard against underperformance by the LLM. Additionally, the student model serves as a reference point for "course tracking," meaning

that it allows us to monitor and guide the annotation process by comparing the student model's output with the teacher model's output. This approach is particularly evident in our experiments where the Meta-8B Instruct model, acting as a low-competency "noisy teacher," demonstrated suboptimal performance on most of the eight datasets, as indicated by its low CAI scores. The student model addresses this issue by collaborating with the teacher model to iteratively refine annotations. This process ensures the framework's robustness, even when the teacher model lacks competency in specific tasks. We justify the necessity of the student model through experimental analysis (see Section 4.3.1 and Table 4). These results show that our proposed Retrospective Learning (ReL) framework consistently outperforms baseline methods, even when paired with low-competency teacher models such as the Llama 8B Instruct model (Touvron et al., 2023). This demonstrates the resilience of ReL and the critical role of the student model in enhancing performance across diverse LLM configurations. Moreover, recent studies (Zhou et al., 2024; Xiong et al., 2023) highlight the inherent challenges of relying solely on LLMs, particularly their tendencies toward overconfidence and reluctance to express uncertainty. These findings further validate the inclusion of a student model to mitigate such limitations.

### A.3 ADDITIONAL EXPERIMENTS TO SUPPORT OUR ARGUMENT

In this section (see Table 13 and Table 14), we evaluate an additional three open-source NLP datasets on ChatGPT-3.5 and ChatGPT-4 Mini, alongside five datasets—Bank77, CLINC (Intent), MTOP (Intent), Massive (Intent), and StackExchange—as well as Reddit (Topic), Few Rel Nat (Type), and Massive Intent (Intent) using Meta-Llama 8B-Instruct. Some modifications were made following (Zhang et al., 2023) to align with the experimental setup.

### A.3.1 EXPERIMENTAL STUDIES ON META-8B INSTRUCT USING RETROSPECTIVE LEARNING

We also conducted an experiment using meta-8B instruct as a "noisy teacher." This model is smaller and less competent than ChatGPT, but we used it with the same student model. This experiment (See Table 14) illustrates the effectiveness of our method and highlights the importance of the student model, even when learning from a much smaller, larger language model. Our proposed Retrospective Learning (RetroL) demonstrates consistent improvement over "Only Student" across all datasets, with the highest improvement observed for StackExchange ($+6.61\%$). Additionally, RetroL outperforms "Only LLMs" across all datasets, highlighting its robustness and effectiveness, particularly in scenarios where the LLM exhibits low competency.

| Datasets | Only Student Model (Our) | Only LLMs (Llama-8B-Instruct) | Student (Our) & LLM (Llama-8B-Instruct) | Student & Teacher KD (Our) | Retrospective Learning (Our)(%) | CAI Ratio & (Before & After) |
|---|---|---|---|---|---|---|
| Clinc | 79.01 ±1.08 | 32.49 ±6.73 | 69.40 ±7.28 | 63.41 ±3.19 | **82.43**±0.20 | 0.56⇒4.43 |
| Massive_Scenario | 75.55 ±1.76 | 43.52 ±1.85 | 66.74 ±0.98 | 70.06±1.12 | **78.13** ±0.74 | 0.67⇒4.88 |
| Mtop Intent | 52.49 ±2.52 | 34.17 ±6.70 | 48.23 ±0.25 | **66.39** ±0.70 | 63.39 ±1.47 | 0.35⇒1.46 |
| StackExchange | 32.27 ±0.65 | 11.02 ±2.78 | 26.26 ±2.16 | 16.03 ±0.13 | **38.88** ±0.27 | 0.23⇒0.53 |
| Banking77 | 73.93 ±1.56 | 33.06 ±1.92 | 69.66 ±1.74 | 64.29 ±1.24 | **77.71** ±0.25 | 0.68⇒4.20 |
| Reddit | 51.73 ±0.62 | 36.31 ±0.97 | 46.00 ±2.51 | 40.29±0.55 | **58.81** ±0.28 | 0.33⇒1.58 |
| Few Rel Nat | 35.35 ±0.016 | 14.25 ±0.36 | 30.07 ±4.45 | 31.80±0.34 | **42.92** ±0.06 | 0.13⇒0.85 |
| Massive_Intent | 61.80 ±1.04 | 45.41 ±0.06 | 56.03 ±0.08 | 67.49 ±0.10 | **67.75** ±0.43 | 0.73⇒2.87 |

Table 13: **Meta-Llama 3-8B Instruct (Open-Source Light-Weight LLMs):** Annotation Accuracy comparison in percentages with standard deviations across different datasets for the Student Model, LLMs without annotations from the student model, and LLMs with annotations from the student model. The highest accuracy for each dataset is highlighted.

| Datasets | Only Student Model (Our) | Only LLMs (Llama-8B-Instruct) | Retrospective Learning (Our) (%) | Improvement Over Only Student (%) | Improvement Over Only LLMs (%) |
|---|---|---|---|---|---|
| Clinc | 79.01 ±1.08 | 32.49 ±6.73 | **82.43** ±0.20 | +3.42 | +49.94 |
| Massive_Scenario | 75.55 ±1.76 | 43.52 ±1.85 | **78.13** ±0.74 | +2.58 | +34.61 |
| Mtop Intent | 52.49 ±2.52 | 34.17 ±6.70 | **63.39** ±1.47 | +10.90 | +29.22 |
| StackExchange | 32.27 ±0.65 | 11.02 ±2.78 | **38.88** ±0.27 | +6.61 | +27.86 |
| Banking77 | 73.93 ±1.56 | 33.06 ±1.92 | **77.71** ±0.25 | +3.78 | +44.65 |
| Reddit | 51.73 ±0.62 | 36.31 ±0.97 | **58.81** ±0.28 | +7.08 | +22.50 |
| Few Rel Nat | 35.35 ±0.016 | 14.25 ±0.36 | **42.92** ±0.06 | +7.57 | +28.67 |
| Massive_Intent | 61.80 ±1.04 | 45.41 ±0.06 | **67.75** ±0.43 | +5.95 | +22.34 |

Table 14: **Performance Comparison of Retrospective Learning (RetroL):** The table shows the performance of Retrospective Learning compared to "Only Student" and "Only LLMs," with improvements highlighted.

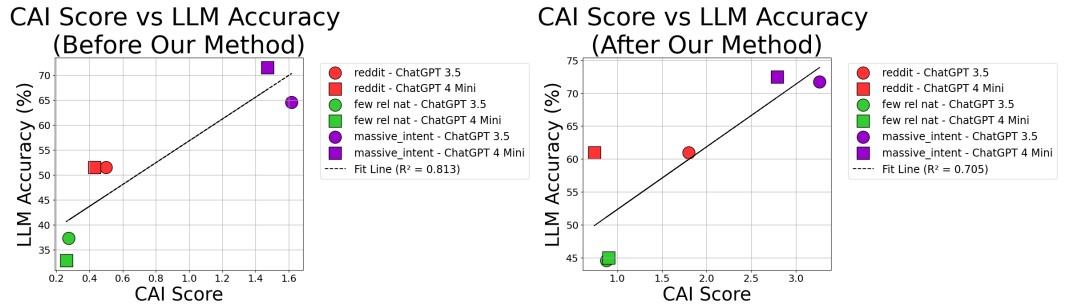

(a: Chatgpt3.5 Turbo and ChatGPT 4o mini)    (b:Chatgpt3.5 Turbo and ChatGPT 4o mini)

Figure 5: The above analysis shows the correlation between LLM annotation accuracy and the Consistent and Inconsistent (CAI) ratio. We also conducted statistical tests to assess the significance of this correlation. We collected the CAI ratios for (LLMs 3.5 Turbo and Student Model) and (LLMs 4.0 Mini and Student Model) across the datasets reddit, few rel nat and massive intent. Using these data, we calculated the Pearson correlation coefficients between the LLM annotation accuracies and CAI ratios and computed the associated P-values (P value for After: 0.03643843400972288) and (P value for Before: 0.014021729786979444) to determine the statistical significance of the observed correlations.

## A.4 COMPARISON OF OUR METHOD WITH LLMS USING PROMPTING TECHNIQUES FOR SELF-CORRECTION

| Metric
Dataset | ChatGPT-4o mini
StackExchange | ChatGPT-4o mini
Clinc | ChatGPT-4o mini
Banking77 | ChatGPT-4o mini
Mote | ChatGPT-4o mini
Massive(D) |
|---|---|---|---|---|---|
| FeedbackShinn et al. (2024) | 51.72% ± 0.27% | 79.34% ± 0.49% | 64.81% ± 1.33% | 71.93% ± 0.02% | 71.35% ± 0.29% |
| CorrectionPaul et al. (2023) | 47.55% ± 0.34% | 81.85% ± 0.63% | 65.58% ± 1.23% | 73.57% ± 0.47% | 70.84% ± 0.05% |
| Retrospective Learning | 45.22% ± 0.15% | 87.93% ± 0.53% | 82.45% ± 0.48% | 67.10% ± 0.32% | 80.18% ± 0.45% |

| Metric
Dataset | ChatGPT-3.5
StackExchange | ChatGPT-3.5
Clinc | ChatGPT-3.5
Banking77 | ChatGPT-3.5
Mote | ChatGPT-3.5
Massive(D) |
|---|---|---|---|---|---|
| FeedbackShinn et al. (2024) | 48.46% ± 0.00% | 71.63% ± 1.24% | 53.90% ± 2.94% | 71.88% ± 0.59% | 63.55% ± 0.02% |
| CorrectionPaul et al. (2023) | 51.81% ± 0.04% | 65.06% ± 0.83% | 55.94% ± 0.32% | 68.24% ± 0.09% | 62.81% ± 0.07% |
| Retrospective Learning | 41.45% ± 2.56% | 85.49% ± 0.19% | 76.92% ± 0.02% | 69.06% ± 1.10% | 76.43% ± 2.47% |

Table 15: The table shows the accuracy results for our methods and LLMs prompting-based baselines evaluated using ChatGPT 3.5 and ChatGPT 4o-o-mini on different datasets.

In this section (See Table 12), we add **Self-Refine**, a method designed to improve initial output through iterative rounds of self-correction (Madaan et al., 2024), and **Reflexion** Shinn et al. (2024), aims to achieve self-correction through LLMs' own evaluations and incorporates feedback from internal or external tools as our additional baselines. Both techniques largely depend on the LLMs its own in handling the corresponding task. These two additional baselines are added to demonstrate that

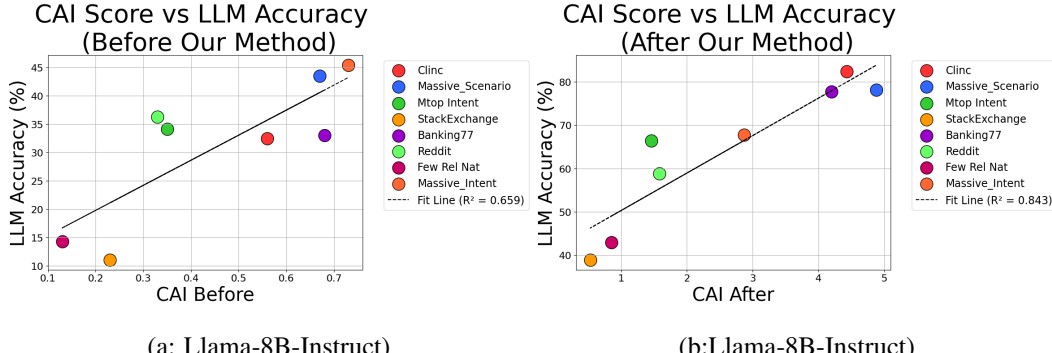

(a: Llama-8B-Instruct)  (b:Llama-8B-Instruct)

Figure 6: The above analysis shows the correlation between LLM annotation accuracy and the Consistent and Inconsistent (CAI) ratio. We also conducted statistical tests to assess the significance of this correlation. We collected the CAI ratios for (LLMs 3.5 Turbo and Student Model) and (LLMs 4.0 Mini and Student Model) across the datasets CLINC, Massive Scenario, MTOP Intent, Stack Exchange, and Banking77, Reddit, Few Rel Nat, Massive_Intent. Using these data, we calculated the Pearson correlation coefficients between the LLM annotation accuracies and CAI ratios and computed the associated P-values (P value for After: 0.014399601133794529) and (P value for Before: 0.0012912893433347605) to determine the statistical significance of the observed correlations.

our retrospective learning which collaborates between a student model and a teacher can outperform LLMs in self-correction for unsupervised datasets with user preferences.

A.5 INVERSE CONSISTENT (IC) RATIO

The number of samples per class required for human annotation based on user preferences is determined by our Inverse Consistent (IC) ratio (7). For user-preference samples. The $n$ denotes the total size of the the consistent sample where $M = n$, and $k$ be the number of classes. The parameter $p$ represents the proportion of samples to be selected and is set to 5% (i.e., $p = 0.05$). In our experiment, we do not use all identified consistent samples. The proportion of consistent samples used for self-correction is determined by the IC ratio. Let $n_c$ be the number of consistent samples, so $M = n_c$ represents the size of the consistent sample selection. If the CAI ratio is greater than 0.5 (i.e., the number of consistent samples exceeds inconsistent ones), the value of $p$ will be reduced to use fewer consistent samples. If the CAI ratio is less than 0.4, $p$ is set to 1 (i.e., 100%) since more consistent samples are needed for self-correction. The formula for the **Inverse Consistent (IC) ratio** is defined as follows:

$$IC = \left( \frac{M \times p}{k} \right). \tag{7}$$

A.6 EXPERIMENTAL DETAILS

The top-k selection and proportions of consistent and user-preference samples are as follows. For CLINC and Massive Scenario, 'top-k' is set to 5, with 'proportion' at 0.2. For MTOP Intent, 'proportion' is set to 1, and 'top-k' is updated to 15 after printing the current value. In StackExchange, 'top-k' is set to 5 and 'proportion' to 1, while in Banking77, 'top-k' is set to 3 and 'proportion' is 0.2. In massive intent, 'top-k' is 20 and 'proportion' is 0.5), proportion=0.2, and few real nat has top-k=30, and proportion is 1. In 'reddit', 'top-k' is set to 7, and the proportion is 0.2. All tests are done with two random seeds with temperature parameters (0.5 and 1) for user preference samples, student model-assigned annotation, and LLMs with and without student annotations.

## A.7 The Prompting Format and Instruction Used for ChatGPT 3.5 and 4o mini

### A.7.1 Prompt Instruction:

```
def Prompt(prompts,student_labels,intention_set,temperatures,formats):
```
1. Initialize an empty list `combination`.
2. For each pair of `prompt` and `student_labels` from `prompts` and `student_labels` lists:
a. Construct `prompt1` as `"For the sentence:   "{prompt}"`.
b. Append `prompt1` to `combination`.
3. Initialize a response string `respon`.
4. Append the following to `respon`:
a. A message ensuring the number of responses corresponds to the length of `prompts`.
b. A request to identify the intentions for each sentence based on `intention_set`.
c. Instructions for response formatting using `formats`.
d. A message to ensure the total responses are as expected from ChatGPT.
5. Use `openai.ChatCompletion.create()` to send the `respon` string, along with temperature and token limits, to the model.
6. Return the model's response as the final output.

### A.8 Comparison of CAI Ratios Before and After Applying Retrospective Learning

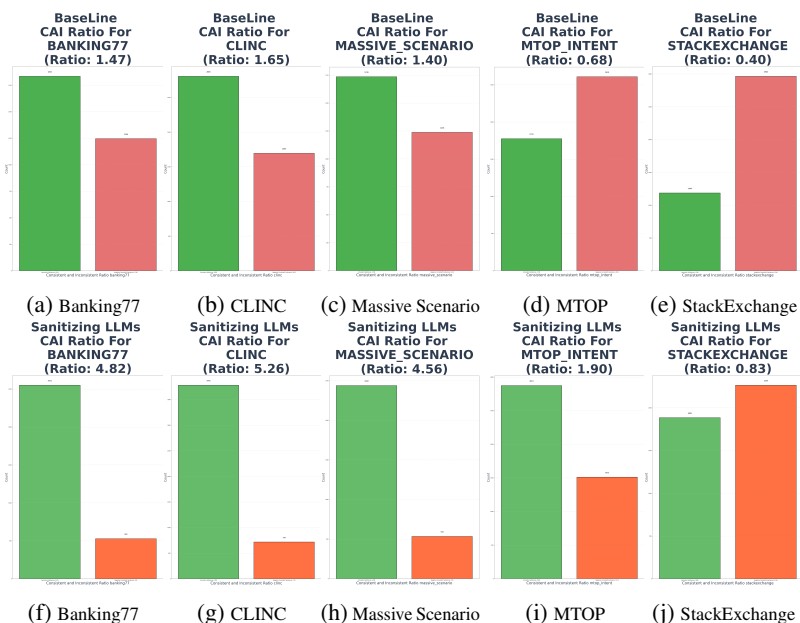

Figure 7: Performance comparison based on LLMs (Chatgpt 3.5) and student model (MINILM (Wang et al., 2020)) across 5 different datasets. The **first row** presents the CAI ratio before applying our sanitizing LLMs for the Student+LLMs (Chatgpt 3.5) baseline, while the **second row** shows the results after applying our proposed sanitizing LLMs, demonstrating a significant reduction in the number of inconsistent samples.

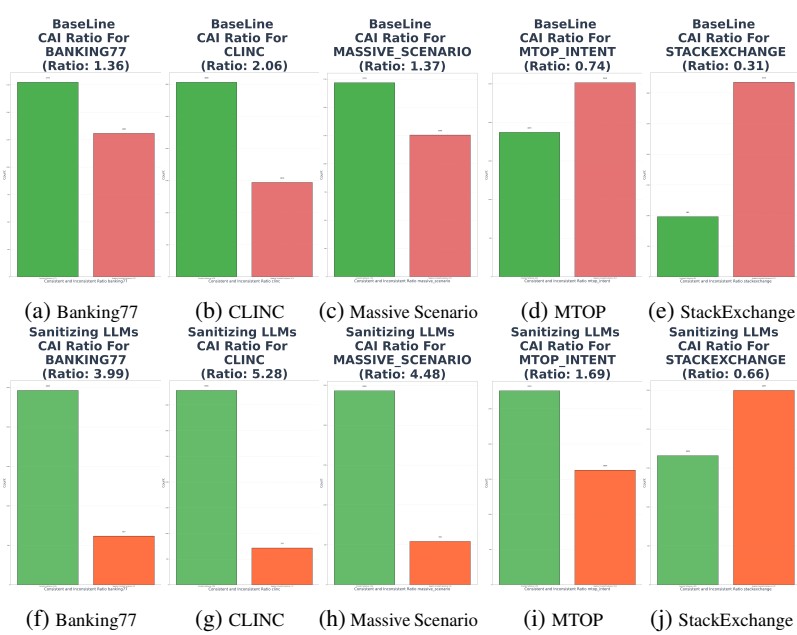

Figure 8: Performance comparison based on LLMs (Chatgpt 4o mini with temperature 1) and student model (MINILM (Wang et al., 2020)) across 5 different datasets. The **first row** presents the CAI ratio before applying our sanitizing LLMs for the Student+LLMs (Chatgpt 3.5) baseline, while the **second row** shows the results after applying our proposed sanitizing LLMs, demonstrating a significant reduction in the number of inconsistent samples.

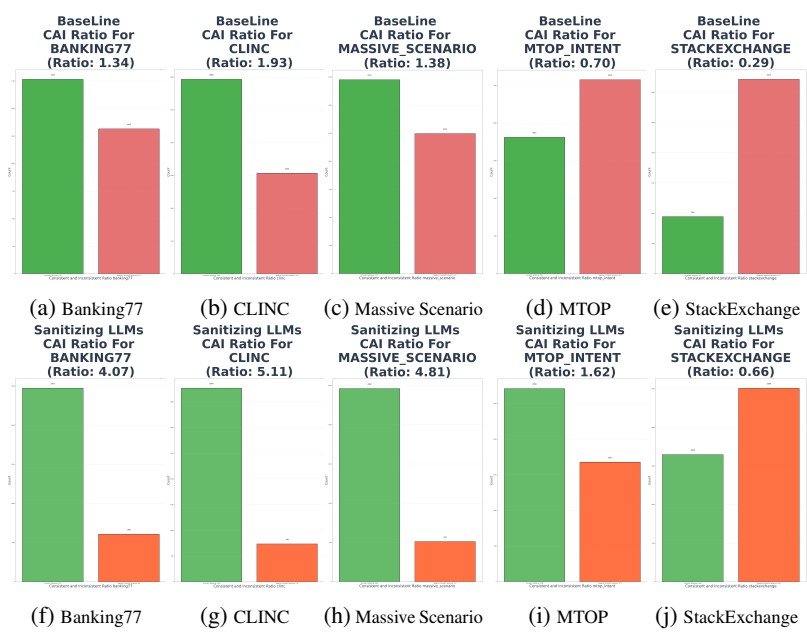

(a) Banking77    (b) CLINC    (c) Massive Scenario    (d) MTOP    (e) StackExchange

(f) Banking77    (g) CLINC    (h) Massive Scenario    (i) MTOP    (j) StackExchange

Figure 9: Performance comparison based on LLMs (Chatgpt 4o mini with temperature 0.5) and student model (MINILM (Wang et al., 2020)) across 5 different datasets. The **first row** presents the CAI ratio before applying our sanitizing LLMs for the Student+LLMs (Chatgpt 3.5) baseline, while the **second row** shows the results after applying our proposed sanitizing LLMs, demonstrating a significant reduction in the number of inconsistent samples.

