# OpenReview forum: "Sanitizing LLMs: Retrospective Learning for Self-Correction of Inconsistent Samples via User Preferences"
_ICLR.cc/2025/Conference — Submitted to ICLR 2025_

### Official Review · Reviewer_vKpA · 2024-11-02

**Soundness:** 2
**Presentation:** 1
**Contribution:** 2
**Rating:** 5
**Confidence:** 2

**Summary:**

Authors propose a metric called Consistent and Inconsistent (CAI) Ratio where they identify the ratio of consistant and inconsistent data points. Then they use this metric and propose a method called Retrospective Learning (RL) to make more inconsistent data points consistant which can help boost the accuracy of the generated data.

**Strengths:**

The approach proposed in the paper is straightforward, easy to understand and easy to implement which can make people to easily use it.

**Weaknesses:**

While the approach proposed in the paper is easy to understand, the paper is not written clearly! Specially the beginning few sections of the paper including the introduction. The reason is that there were many undefined terms in the beginning of the paper that would make it hard to understand the paper in the beginning. The reader might understand the paper only after going through the paper multiple times and decoding the terms. The paper might need significant revision to make it clear. In addition, there are confusing terminology used in the paper like using RL for Retrospective Learning. I would advise changing these acronyms to something that is not widely used to avoid confusion. In addition, the motivation of the work is not super clear to me either. All these points resulted in the paper to not be fully clear and made it hard for me to assess the paper completely.

Another concern is that authors use the same metric as an evaluation metric (CAI) based on which the mitigation approach is proposed which might not be fair. Although authors use accuracy as a metric as well, it would be good to use other metrics to make the results more robust. Can authors do some human experiments or perhaps use other metrics as well?

**Questions:**

Mentioned the question in the weaknesses section above.

---

> ### Author Response · Authors · 2024-11-25
>
> **While the approach proposed in the paper is easy to understand, the paper is not written clearly! Specially the beginning few sections of the paper including the introduction. The reason is that there were many undefined terms in the beginning of the paper that would make it hard to understand the paper in the beginning.**
>
>
> Dear Reviewer,
> Thank you very much for your detailed observations and valuable feedback. We truly appreciate your time and effort in reviewing our work.
> We have carefully reviewed the introduction and early sections of the paper and added more clarifications for some of the introduced terms that may have been unclear. These updates are now reflected in the main paper. Please let us know if there are still terms or concepts that are confusing, and we will promptly address them in the main paper.
>
>
> **In addition, there are confusing terminology used in the paper like using RL for Retrospective Learning. I would advise changing these acronyms to something that is not widely used to avoid confusion.**
>
> Regarding the use of "RL" for Retrospective Learning, we appreciate your suggestion to avoid potential confusion with widely used acronyms. To improve clarity, we have updated the abbreviation to **RetroL** throughout the paper.
> Thank you once again for your helpful suggestions, which have significantly improved the readability and clarity of our paper.

---

> ### Author Response · Authors · 2024-11-25
>
> **Comment 2: Can authors do some human experiments or perhaps use other metrics as well ?**
>
> Dear Reviewer,
> For the test, we already have the ground truth that can serve as a valid metric to evaluate our proposed method. We have the ground truth, for testing. Thus, there is no need to have human experiments. In contrast, we have added additional datasets to show the effectiveness of our proposed metric and method. Lastly, adding human experiments can be further studied in our future work.
>
> Thank you for your question regarding the use of other metrics. Accuracy is indeed a crucial metric for evaluating our retrospective learning, as it provides an objective measure of how well our method performs self-correction. The CAI ratio can be particularly useful in real-world scenarios with limited or no available annotations. Moreover, it is particularly useful given that we do not know the nature of LLMs for a specific task, especially in unsupervised data with user preference. It helps determine the reliability and trustworthiness of annotations generated by LLMs in such situations. Our experiments in Table 2, 3,4 in the main paper have proved that.

---

> > ### Author Response · Authors · 2024-11-25
> >
> > **Comment 3: In addition, the motivation of the work is not super clear to me either.**
> >
> > Dear Reviewer,
> > Thank you for your question regarding the motivation of our work. Let me clarify this further.
> > In unsupervised learning tasks that rely on user preferences to align data annotations with expectations—where the competency of the teacher model (LLMs) is uncertain and no external knowledge is available—the key challenge lies in evaluating the annotations generated by LLMs and enabling mechanisms for self-correction. To address this, we propose a novel approach termed Retrospective Learning (ReL), a self-supervised framework designed to facilitate both self-correction and self-assessment of annotations produced by LLMs.
> >
> > Our methodology leverages a student model to collaborate with a teacher model of uncertain competency. By introducing the Consistent and Inconsistent (CAI) score and (CAI) identification, we systematically quantify and identify consistent and inconsistent samples. This iterative refinement process enhances the performance of both the student and teacher models.
> > To demonstrate the efficacy of our approach, we conducted experiments using the Meta-8B Instruct lightweight LLM as a low-competency “noisy teacher.” This setup, combined with the student model, underscores the robustness of our framework and highlights the pivotal role of the student model in managing scenarios involving noisy teachers.
> > In summary, given the challenges of unsupervised learning tasks—where the teacher’s competency is unknown and no external knowledge is available—we aim to answer: How can we evaluate and enable self-correction for an unsupervised dataset? Our proposed Retrospective Learning strategy addresses this by enabling both self-correction and self-evaluation of LLM-generated annotations. Through collaboration between the student and teacher models, the CAI score helps identify consistent and inconsistent samples, ultimately improving the performance of both models.
> >
> > Thank you again for your question, and we hope this explanation clarifies the motivation of our work.

---

> > > ### Author Response · Authors · 2024-12-02
> > > **Have we addressed your concerns ?**
> > >
> > > Dear Reviewer vKpA:
> > >
> > > Thank you again for your time and valuable feedback. As the rebuttal period is nearing its end, we hope our response has addressed your concerns. If so, we kindly request that you update your review and raise your score accordingly. Please let us know if you have any further questions or comments.
> > >
> > > Thank you!

---

> > > ### Author Response · Authors · 2024-12-04
> > > **Additional Information**
> > >
> > > **Dear Reviewer vKpA,**
> > >
> > > **Comment 3: In addition, the motivation of the work is not super clear to me either.**
> > >
> > > One of the key motivations for our approach is its efficiency, which enables significant improvements in annotation accuracy while minimizing reliance on the teacher model. This efficiency manifests in two important ways:
> > >
> > > 1. **Computational Efficiency**: Our method requires access to the teacher model only twice per dataset, with little (one shot) to no (zero-shot) reliance on demonstrations for prompting. This reduces computational overhead significantly.
> > >
> > > 2. **Cost-Effectiveness**: For closed-source models with API service fees, our approach minimizes costs. By leveraging the student model in conjunction with our proposed clustering operation and limited teacher model predictions, our method achieves superior performance compared to both models, at a lower cost. This is particularly advantageous when contrasted with approaches that rely on iterative self-correction and feedback loops.
> > >
> > > We hope this clarification highlights the practical advantages of our approach.
> > >
> > > Thank you !

---

### Official Review · Reviewer_d96A · 2024-11-04

**Soundness:** 3
**Presentation:** 3
**Contribution:** 2
**Rating:** 6
**Confidence:** 3

**Summary:**

For ensure the model generated labels/responses are aligned with end-user needs, this paper first proposes a consistent and inconsistent ratio for evaluating the performance of LLMs or student generated annotations. The main motivation is the consistent samples have significantly higher accuracy than inconsistent ones. This ratio is used to identify consistent and inconsistent samples. For resolving the self-correction with limited user preference data, this paper also introduces a new method retrospective learning to identify and correct inconsistent samples for minimizing inconsistency and enhancing the accuracy of LLM annotations. The proposed method achieves a better classification accuracy consistently cross datasets with a higher consistent and inconsistent ratio.

**Strengths:**

Strengths:

1.	For assessing the annotations generated by LLMs or student model, this paper proposes a consistent and inconsistent sample identification and ratio. It can identify the consistent and inconsistent samples. It utilized the predictions from teacher model and student model. If the predictions from these two models are same, then it is a consistent sample. Vise versa.

2.	The retrospective learning method is proposed for self-correction. It includes two modules: divide-and-conquer self-correction and majority voting. With this method, the prediction accuracy is improved because inconsistent samples impact the model performance.

**Weaknesses:**

Weaknesses:

1.	Although this paper has shown some results on various datasets, it still lacks the comparison with other methods that can handle the similar problem.

2.	More experiments and explanations about the reliability of the CAI score is needed. And this score is highly depended on the selected teacher model and student model. More explorations and comparisons are needed here.

**Questions:**

Please check the weaknesses.

---

> ### Author Response · Authors · 2024-11-25
>
> Dear Reviewer
> Thank you for your constructive feedback and comments. To address them, we have prepared the following responses.
>
> **Comment.1: Although this paper has shown some results on various datasets, it still lacks the comparison with other methods that can handle the similar problem.**
>
> **The following is in response to Comment 1.**
> we have added **Self-Refine**, a method designed to improve initial output through iterative rounds of self-correction [1]  and **Reflexion** [2]  aims to achieve self-correction through LLMs' own evaluations and incorporates feedback from internal or external tools as our additional baselines. Both techniques largely depend on the LLMs on their own in handling the corresponding task.
> By including these baselines, we aim to demonstrate that our proposed **Retrospective Learning** framework, which facilitates collaboration between a student model and a teacher model, outperforms LLMs in self-correction for unsupervised datasets with user preferences. The comparison highlights the benefits of leveraging the synergy between student and teacher models, which is absent in methods that depend solely on LLMs for self-correction.
>
>
> ### Table: Accuracy Results for Methods and LLM Prompting-Based Baselines
>
> #### **ChatGPT-4o Mini**
> | Metric                           | StackExchange         | Clinc               | Banking77           | Mote                | Massive(D)         |
> |----------------------------------|-----------------------|---------------------|---------------------|---------------------|--------------------|
> | Feedback [Shinn et al., 2024]    | **51.72% ± 0.27%**    | 79.34% ± 0.49%      | 64.81% ± 1.33%      | 71.93% ± 0.02%      | 71.35% ± 0.29%     |
> | Correction [Paul et al., 2023]   | 47.55% ± 0.34%        | 81.85% ± 0.63%      | 65.58% ± 1.23%      | **73.57% ± 0.47%**  | 70.84% ± 0.05%     |
> | Retrospective Learning           | 45.22% ± 0.15%        | **87.93% ± 0.53%**  | **82.45% ± 0.48%**  | 67.10% ± 0.32%      | **80.18% ± 0.45%** |
>
> #### **ChatGPT-3.5**
> | Metric                           | StackExchange         | Clinc               | Banking77           | Mote                | Massive(D)         |
> |----------------------------------|-----------------------|---------------------|---------------------|---------------------|--------------------|
> | Feedback [Shinn et al., 2024]    | 48.46% ± 0.00%        | 71.63% ± 1.24%      | 53.90% ± 2.94%      | **71.88% ± 0.59%**  | 63.55% ± 0.02%     |
> | Correction [Paul et al., 2023]   | **51.81% ± 0.04%**    | 65.06% ± 0.83%      | 55.94% ± 0.32%      | 68.24% ± 0.09%      | 62.81% ± 0.07%     |
> | Retrospective Learning           | 41.45% ± 2.56%        | **85.49% ± 0.19%**  | **76.92% ± 0.02%**  | 69.06% ± 1.10%      | **76.43% ± 2.47%** |
> **Notes**:
> - The table compares accuracy results for ChatGPT-3.5 and ChatGPT-4o Mini on various datasets: StackExchange, Clinc, Banking77, Mote, and Massive(D).

---

> > ### Author Response · Authors · 2024-11-25
> >
> > **Dear Reviewer,**
> >
> > Thank you for your insightful feedback regarding the reliability of the CAI score and its dependency on the teacher and student models. To address your concerns, we have expanded our experimental study to include three additional NLP datasets, bringing the total to eight: **Bank77, CLINC (Intent), MTOP (Intent), Massive (Intent), StackExchange, Reddit (Topic), Few Rel Nat (Type), and Massive Intent (Intent)**. We have also incorporated an open-source lightweight LLM (**Meta-Llama 8B Instruct**) alongside two closed-source LLMs (**ChatGPT-3.5 Turbo** and **ChatGPT-4 Mini**), increasing the total number of evaluated models to three.
> > Our study evaluates the reliability of the CAI score and the effectiveness of our proposed Retrospective Learning (ReL) framework across these diverse LLMs and datasets. The updated results, presented in the main paper (Section 4.3, *Experimental Results*) and appendix (Sections A.2 and A.3), demonstrate the robustness of the CAI score in identifying LLM performance in unsupervised tasks with user preferences.
> >
> > ### Results Summary
> > **1. Pearson Correlation Analysis for ChatGPT-3.5 Turbo and ChatGPT-4 Mini (3 Additional Datasets):**
> > | Metric | Pearson Correlation Coefficient | p-value |
> > |---------------------------------|----------------------------------|---------|
> > | CAI_Before vs Accuracy_Before | 0.9017 | 0.0140 |
> > | CAI_After vs Accuracy_After | 0.8398 | 0.0364 |
> > These results indicate strong positive correlations, with statistically significant p-values (p < 0.05).
> > **2. Pearson Correlation Analysis for Meta-Llama 8B Instruct (8 Datasets):**
> > | Metric | Pearson Correlation Coefficient | p-value |
> > |---------------------------------|----------------------------------|---------|
> > | CAI_Before vs Accuracy_Before | 0.8118 | 0.0144 |
> > | CAI_After vs Accuracy_After | 0.9181 | 0.0013 |
> > Similarly, these results show strong positive correlations, with highly significant p-values (p < 0.05).
> > ### Key Observations
> >
> > - **Effectiveness of CAI Score:** The CAI score consistently demonstrates a strong positive correlation with accuracy across all tested models and datasets, validating its reliability in evaluating LLM performance.
> > - **Performance Discrepancy Between LLMs:** Meta-8B Instruct consistently achieves lower CAI scores compared to ChatGPT-3.5 and ChatGPT-4 Mini, aligning with its lower average accuracy across datasets. This highlights the CAI score's ability to reflect model performance and reliability effectively.
> >
> > Our CAI score has proven effective across both closed-source LLMs (ChatGPT-3.5 Turbo and ChatGPT-4 Mini) and open-source lightweight LLMs (Meta-Llama 8B Instruct), as evidenced by statistically significant results from the Pearson correlation analysis.
> > We have updated the main paper to include these new results (see Section 4.3, *Experimental Results*), along with detailed statistical analyses in the appendix (see Section A.2.1, *Statistical Inference*).
> > We hope these expanded experiments and additional explanations address your concerns. Thank you again for your valuable feedback, which has significantly improved the scope and depth of our work.

---

> ### Author Response · Authors · 2024-11-25
>
> ### Table: ChatGPT-3.5 Turbo (Closed-source LLMs)
>
> | **Datasets**          | **Only Student Model (Our)** | **Only LLMs (ChatGPT 3.5)** | **Student + LLM (ChatGPT 3.5)** | **Clustering-Based Method [Zhang et al.]** | **Student & Teacher KD (Our)** | **Retrospective Learning (Our) (%)** | **CAI Score (Before → After)** |
> |------------------------|-----------------------------|-----------------------------|---------------------------------|------------------------------------------|--------------------------------|---------------------------------------|----------------------------------|
> | **Clinc**             | 79.01                      | 66.58                      | 76.82                          | 78.58                                   | 81.32                          | **85.49**                           | 1.55                             |
> | *Std Dev*             | ±1.08                      | ±3.36                      | ±1.51                          | ±0.41                                   | ±0.46                          | ±0.19                               | 5.50                             |
> | **Massive_Scenario**   | 75.55                      | 60.89                      | 70.23                          | 60.85                                   | 69.25                          | **76.43**                           | 1.39                             |
> | *Std Dev*             | ±1.76                      | ±0.62                      | ±1.64                          | ±4.33                                   | ±0.03                          | ±2.47                               | 4.72                             |
> | **Mtop Intent**        | 52.49                      | 64.95                      | 55.12                          | 37.22                                   | **79.57**                      | 69.06                              | 0.68                             |
> | *Std Dev*             | ±2.52                      | ±0.21                      | ±3.08                          | ±1.18                                   | ±0.42                          | ±1.10                               | 1.78                             |
> | **StackExchange**      | 32.27                      | 30.10                      | 30.92                          | **47.75**                              | 29.76                          | 41.45                              | 0.40                             |
> | *Std Dev*             | ±0.65                      | ±0.10                      | ±2.21                          | ±1.24                                   | ±0.19                          | ±2.56                               | 0.85                             |
> | **Banking77**          | 73.93                      | 60.29                      | 73.15                          | 71.20                                   | 70.11                          | **76.92**                           | 1.46                             |
> | *Std Dev*             | ±0.81                      | ±1.33                      | ±1.70                          | ±1.59                                   | ±0.12                          | ±0.02                               | 4.91                             |
> | **Reddit**             | 51.73                      | 51.12                      | 51.64                          | 57.02                                   | 43.90                          | **58.77**                           | 0.50                             |
> | *Std Dev*             | ±0.62                      | ±1.27                      | ±0.18                          | ±1.59                                   | ±1.59                          | ±0.29                               | 1.40                             |
> | **Few Rel Nat**        | 35.35                      | 32.87                      | 37.37                          | **51.22**                              | 49.24                          | 44.88                              | 0.28                             |
> | *Std Dev*             | ±0.016                     | ±1.72                      | ±0.13                          | ±1.43                                   | ±0.63                          | ±0.05                               | 0.89                             |
> | **Massive_Intent**     | 61.80                      | 71.52                      | 64.54                          | 60.69                                   | 73.41                          | **71.72**                           | 1.62                             |
> | *Std Dev*             | ±1.04                      | ±0.95                      | ±0.024                         | ±0.024                                  | ±1.843                         | ±0.40                               | 2.81                             |

---

> > ### Author Response · Authors · 2024-11-25
> >
> > ### Table: ChatGPT-4o Mini (Closed-source LLMs)
> >
> > | **Datasets**          | **Only Student Model (Our)** | **Only LLMs (ChatGPT-4o Mini)** | **Student + LLM (ChatGPT-4o Mini)** | **Clustering-Based Method [Zhang et al.]** | **Student & Teacher KD (Our)** | **Retrospective Learning (Our) (%)** | **CAI Score (Before → After)** |
> > |------------------------|-----------------------------|----------------------------------|-------------------------------------|------------------------------------------|--------------------------------|---------------------------------------|----------------------------------|
> > | **Clinc**             | 79.01                      | 81.44                           | 78.58                              | 78.58                                   | 85.23                          | **87.93**                           | 2.06                             |
> > | *Std Dev*             | ±1.08                      | ±0.44                           | ±1.35                              | ±0.41                                   | ±0.98                          | ±0.53                               | 5.20                             |
> > | **Massive_Scenario**   | 75.55                      | 66.83                           | 77.62                              | 60.85                                   | 79.60                          | **80.18**                           | 1.37                             |
> > | *Std Dev*             | ±1.76                      | ±1.31                           | ±0.74                              | ±4.33                                   | ±0.85                          | ±0.45                               | 4.65                             |
> > | **Mtop Intent**        | 52.49                      | 75.03                           | 57.01                              | 37.22                                   | **80.16**                      | 67.10                              | 0.74                             |
> > | *Std Dev*             | ±2.52                      | ±1.35                           | ±0.37                              | ±1.18                                   | ±0.85                          | ±0.32                               | 1.66                             |
> > | **StackExchange**      | 32.27                      | **51.90**                       | 45.49                              | 47.75                                   | 35.63                          | 45.22                              | 0.31                             |
> > | *Std Dev*             | ±0.65                      | ±0.75                           | ±0.94                              | ±1.24                                   | ±0.51                          | ±0.15                               | 0.66                             |
> > | **Banking77**          | 73.93                      | 65.12                           | 75.39                              | 71.20                                   | 73.56                          | **82.45**                           | 1.36                             |
> > | *Std Dev*             | ±1.56                      | ±0.30                           | ±0.32                              | ±1.59                                   | ±0.20                          | ±0.48                               | 4.03                             |
> > | **Reddit**             | 51.73                      | 53.25                           | 57.40                              | 57.02                                   | 44.47                          | **60.94**                           | 0.51                             |
> > | *Std Dev*             | ±0.62                      | ±0.35                           | ±1.96                              | ±1.59                                   | ±0.69                          | ±0.11                               | 1.90                             |
> > | **Few Rel Nat**        | 35.35                      | 37.11                           | 38.87                              | **51.22**                              | 49.53                          | 44.94                              | 0.26                             |
> > | *Std Dev*             | ±0.016                     | ±0.03                           | ±1.88                              | ±1.43                                   | ±0.35                          | ±0.02                               | 0.90                             |
> > | **Massive_Intent**     | 61.80                      | 66.02                           | 76.93                              | 60.69                                   | **78.93**                      | 72.49                              | 1.47                             |
> > | *Std Dev*             | ±1.04                      | ±0.35                           | ±1.05                              | ±0.024                                  | ±0.50                          | ±0.40                               | 3.30                             |

---

> > > ### Comment · Reviewer_d96A · 2024-12-02
> > >
> > > Thanks a lot for authors' response and additional experiments to address my concerns. The new results further proved the proposed method can achieve a reasonable performance cross many tasks. In addition, it is very important to verify the reliability of the CAI score from authors' response. After considering all aspects, I'd like to raise my score.

---

> > > > ### Author Response · Authors · 2024-12-02
> > > >
> > > > Dear Reviewer d96A:
> > > >
> > > > Thank you very much for your thoughtful feedback and for raising your score. We sincerely appreciate your recognition of our efforts to address the major concerns. Your suggestions are invaluable in helping us improve the paper, and we will thoroughly revise it to incorporate all of your feedback.
> > > >
> > > > Thank you!

---

> > > ### Author Response · Authors · 2024-12-02
> > > **Have we address your concerns ?**
> > >
> > > Dear Reviewer d96A:
> > > Thank you again for your time and valuable feedback. As the rebuttal period is nearing its end, we hope our response has addressed your concerns. If so, we kindly request that you update your review and raise your score accordingly. Please let us know if you have any further questions or comments.
> > >
> > > Thank you!

---

### Official Review · Reviewer_4bru · 2024-11-05

**Soundness:** 2
**Presentation:** 2
**Contribution:** 2
**Rating:** 5
**Confidence:** 3

**Summary:**

This paper introduces the Consistent and Inconsistent (CAI) Ratio which serves as an effective evaluation metric for unsupervised data with user preferences and Retrospective Learning(RL) to self-correct the identified inconsistent samples using consistent samples.

**Strengths:**

- The results are diverse and demonstrate the effectiveness of the proposed approach.
- The idea behind the proposed ratio and data-centric approach is intuitive.

**Weaknesses:**

- Figure 1 lacks clarity and should present each circle in sequential order. Additionally, the notation in the text does not correspond with the notation in the figure.
- In fact, my primary question throughout the entire paper is why a student model is necessary at all; this aspect does not seem to be addressed in the introduction. The motivation seems to lack clarity and justification.
- There is a lack of detail regarding the clustering operations.

**Questions:**

- At line 116, referencing a work from 1999 cannot be considered "recently."
- Additionally, careful attention should be given to the appropriate use of \citet and \citep for citations, as well as ensuring consistent and correct application of \ref and \eqref.

---

> ### Author Response · Authors · 2024-11-25
>
> Dear Reviewer
>
> Thank you for your constructive feedback and comments. To address them, we have prepared the following responses. We have clarified the proposed retrospective learning through extensive experiments. These are detailed below:
>
> **Comment.1: Figure 1 lacks clarity and should present each circle in sequential order. Additionally, the notation in the text does not correspond with the notation in the figure.**
>
> Dear Reviewer,
> Thank you for your insightful feedback on Figure 1. We acknowledge the issues with clarity, sequential presentation, and inconsistent notation. We have updated Figure 1 in the main paper to address these concerns, ensuring that each circle is presented in sequential order and that the notation in the text aligns with the figure. Numbering has been added to illustrate the self-correction process clearly and sequentially.

---

> ### Author Response · Authors · 2024-11-25
>
> **Comment.2: In fact, my primary question throughout the entire paper is why a student model is necessary at all; this aspect does not seem to be addressed in the introduction. The motivation seems to lack clarity and justification.**
>
>
> The inclusion of the student model is essential as it provides a safeguard against underperformance by the LLM. *Additionally, the student model serves as a reference point for "course tracking," meaning that it allows us to monitor and guide the annotation process by comparing the student model's output with the teacher model's output.* This is particularly evident in our experiments where the Meta-8B Instruct model, serving as a low-competency "noisy teacher," exhibited suboptimal performance on most of the eight datasets, as reflected in its low CAI scores. The student model addresses this challenge by collaborating with the teacher model to iteratively refine annotations, ensuring robustness even when the teacher model lacks competency in specific tasks. We justify the necessity of the student model through experimental analysis (see Section 4.3.1 and Table 4 for more details).
>
> These results demonstrate  (in the following table ) that our proposed Retrospective Learning (ReL) framework consistently outperforms baselines, even when deployed with low-competency teacher models like the Llama 8B Instruct model [3]. This highlights the resilience of ReL and the critical role of the student model in improving performance across diverse LLM configurations.
> Furthermore, recent studies [1, 2] underscore the inherent challenges in relying solely on LLMs due to overconfidence and reluctance to express uncertainty, which further validates the need for a student model to mitigate these limitations.
> Thank you again for pointing this out, and we hope this explanation addresses your concern.
>
> **Comment.3: There is a lack of detail regarding the clustering operations.**
>
> Thank you for your valuable feedback. We will revise the manuscript to include more details about the clustering operations. We believe this will address your concern and clarify our approach. However, we would like to note that text clustering is not the primary focus of our work.
>
> **Remaining Questions:**
>
> *Response to "At line 116, referencing a work from 1999 cannot be considered 'recently'.":*
>
> Thank you for pointing this out. You are absolutely correct. We have updated the reference to a more recent work in the main paper to ensure the information is current and relevant. We apologize for this oversight.
>
> *Response to "Additionally, careful attention should be given to the appropriate use of \citet and \citep for citations, as well as ensuring consistent and correct application of \ref and \eqref."*
>
> Thank you for bringing this to our attention. We have carefully reviewed the manuscript and revised the use of \citet and \citep to ensure they are used consistently and correctly according to our chosen citation style. We have also verified the application of \ref and \eqref for accurate cross-referencing within the document. We appreciate your feedback, as it helped us improve the overall quality and consistency of the paper.
>
>
> Reference:
>
> [1] Can LLMs Express Their Uncertainty? An Empirical Evaluation of Confidence Elicitation in LLMs
> Miao Xiong, Zhiyuan Hu, Xinyang Lu, YIFEI LI, Jie Fu, Junxian He, Bryan Hooi
>
> [2] Kaitlyn Zhou, Jena D Hwang, Xiang Ren, and Maarten Sap. 2024b. Relying on the unreliable: The impact of language models’ reluctance to express uncertainty
>
> [3] Touvron, H., Lavril, T., Izacard, G., Martinet, X., Lachaux, M.A., Lacroix, T., Rozière, B., Goyal, N., Hambro, E., Azhar, F. and Rodriguez, A., 2023. Llama: Open and efficient foundation language models.

---

> > ### Author Response · Authors · 2024-11-25
> >
> > ### Table: Meta-Llama 3-8B Instruct (Open-Source Light-Weight LLMs)
> >
> > **Annotation Accuracy comparison in percentages with standard deviations across different datasets for the Student Model, LLMs without annotations from the Student Model, and LLMs with annotations from the Student Model. The highest accuracy for each dataset is highlighted.**
> >
> > | **Datasets**        | **Only Student Model (Our)** | **Only LLMs (Llama-8B-Instruct)** | **Student (Our) & LLM (Llama-8B-Instruct)** | **Student & Teacher KD (Our)** | **Retrospective Learning (Our) (%)** | **CAI Score (Before → After)** |
> > |----------------------|-----------------------------|-----------------------------------|--------------------------------------------|--------------------------------|--------------------------------------|----------------------------------|
> > | **Clinc**           | 79.01 ±1.08                | 32.49 ±6.73                      | 69.40 ±7.28                                 | 63.41 ±3.19                   | **82.43 ±0.20**                     | 0.56 → 4.43                     |
> > | **Massive_Scenario**| 75.55 ±1.76                | 43.52 ±1.85                      | 66.74 ±0.98                                 | 70.06 ±1.12                   | **78.13 ±0.74**                     | 0.67 → 4.88                     |
> > | **Mtop Intent**     | 52.49 ±2.52                | 34.17 ±6.70                      | 48.23 ±0.25                                 | **66.39 ±0.70**               | 63.39 ±1.47                         | 0.35 → 1.46                     |
> > | **StackExchange**   | 32.27 ±0.65                | 11.02 ±2.78                      | 26.26 ±2.16                                 | 16.03 ±0.13                   | **38.88 ±0.27**                     | 0.23 → 0.53                     |
> > | **Banking77**       | 73.93 ±1.56                | 33.06 ±1.92                      | 69.66 ±1.74                                 | 64.29 ±1.24                   | **77.71 ±0.25**                     | 0.68 → 4.20                     |
> > | **Reddit**          | 51.73 ±0.62                | 36.31 ±0.97                      | 46.00 ±2.51                                 | 40.29 ±0.55                   | **58.81 ±0.28**                     | 0.33 → 1.58                     |
> > | **Few Rel Nat**     | 35.35 ±0.016               | 14.25 ±0.36                      | 30.07 ±4.45                                 | 31.80 ±0.34                   | **42.92 ±0.06**                     | 0.13 → 0.85                     |
> > | **Massive_Intent**  | 61.80 ±1.04                | 45.41 ±0.06                      | 56.03 ±0.08                                 | 67.49 ±0.10                   | **67.75 ±0.43**                     | 0.73 → 2.87                     |

---

> > ### Author Response · Authors · 2024-11-26
> > **Comment.3: There is a lack of detail regarding the clustering operations.**
> >
> > Dear Reviewer
> >
> > We have added a more detailed explanation regarding the clustering operation in the following:
> >
> > ### Clustering Operations in Retrospective Learning
> >
> > The clustering operation is conducted using a semantic similarity score equation and majority voting based on a top-nearest embedding scheme. This process is crucial in our retrospective learning framework, aligning annotations with user-defined preferences. By assigning annotations in this manner, the **Consistent Annotation Identification (CAI)** method facilitates self-correction of identified inconsistent samples through the Divide-and-Conquer Self-Correction (DCSC) process—an iterative self-correction mechanism. Our approach supports user-defined preferences and is applicable to a wide range of large NLP datasets, making it particularly impactful for unsupervised tasks involving user preferences.
> >
> > ### Steps of the Clustering Operation:
> >
> > 1.  **Extracting Embeddings from the Student Model**
> >
> >     Semantic features of text inputs are obtained using a pretrained student model (e.g., MiniLM) to generate dense vector representations. For a given sample $x_i$, the embedding function $S(x_i)$ computes an embedding $e_i$, capturing the sample's semantic features.
> > 2.  **User-Preference Sample Clustering**
> >
> >     A small annotated set of user-preference samples $H = \{x_1, \dots, x_s\}$ is provided. These samples are partitioned into $k$ clusters, denoted as $C_1, C_2, \dots, C_k$, based on predefined labels $y_j \in \mathcal{Y}$, which reflect the user’s annotation preferences. The clusters satisfy the conditions:
> >
> >     $C_i \cap C_j = \emptyset, \quad \forall i \neq j, \quad \text{and} \quad \bigcup_{i=1}^k C_i = H$.
> > 3.  **Annotation Assignment for Unlabeled Data Using the Student Model**
> >
> >     For each unlabeled sample $x_i \in D_u$, its embedding $e_i = S(x_i)$ is computed. The semantic similarity of $e_i$ to each cluster $C_j$ is calculated using the **Average Similarity (AS)**:
> >
> >     $AS(e_i, C_j) = \frac{1}{k} \sum_{e \in \text{Top-}k(C_j, e_i)} \frac{e_i \cdot e}{\|e_i\| \|e\|}$,
> >
> >     where $e$ represents the embeddings in $C_j$, and $\text{Top-}k(C_j, e_i)$ is the subset of samples in $C_j$ with the top $k$ cosine similarity scores to $e_i$. The cluster $C_{j^*}$ with the highest $AS(e_i, C_j)$ is selected, and its label $\bar{y}_{j^*}$ is assigned to $x_i$. The resulting student-annotated dataset is denoted as $D_s = \{(x_i, \bar{y}_i)\}$.
> > 4.  **Annotation Assignment for Unlabeled Data Using the Teacher Model (LLMs)**
> >
> >     Using $D_s$, annotations are refined with a teacher model (LLMs) via two prompting strategies:
> >
> >     *   **Zero-shot prompting:** No annotations from the student model are included.
> >     *   **Single-shot prompting:** Student annotations are incorporated.
> >
> > 5.  **Identification of Consistent and Inconsistent Samples**
> >
> >     In the absence of ground truth annotations, we introduce the **CAI score**, a metric for evaluating annotation quality in unsupervised tasks. This score identifies consistent and inconsistent samples by comparing the annotations from the student and teacher models. Further details on the CAI score and its calculation are provided in Section 3.2 of the main paper.
> >
> > 6.  **Majority Voting via Top-Nearest Embedding Scheme (MV-VTES) and Divide-and-Conquer Self-Correction (DCSC)**
> >
> >     To correct corrupted annotations in inconsistent samples, we propose MV-VTES and the DCSC framework.
> >
> >     *   **MV-VTES:** Uses consistent samples and user-preference samples to iteratively refine annotations for inconsistent samples based on majority voting within the top-nearest embeddings.
> >     *   **DCSC:** A divide-and-conquer mechanism that iteratively corrects annotations for identified inconsistent samples.
> >
> >     Consistent samples, having significantly higher accuracy, serve as a reliable reference for annotation refinement. For detailed algorithms, refer to Section 3.3 of the main paper.

---

> > ### Author Response · Authors · 2024-12-04
> > **Additional Information**
> >
> > **Comment.2: In fact, my primary question throughout the entire paper is why a student model is necessary at all; this aspect does not seem to be addressed in the introduction. The motivation seems to lack clarity and justification.**
> >
> > **Dear Reviewer 4bru,**
> >
> > Thank you for your feedback. One of the key motivations for our approach, including the use of a student model, is to enhance efficiency. This efficiency is evident in two significant aspects:
> >
> > 1. **Computational Efficiency**: Our method requires access to the teacher model only twice per dataset, with minimal or no reliance on demonstrations for prompting. This substantially reduces computational overhead.
> >
> > 2. **Cost-Effectiveness**: For closed-source models with API service fees, our approach offers a cost-efficient solution. By utilizing the student model alongside our proposed clustering operation and a limited number of teacher model predictions, our method achieves superior performance compared to both models individually. Importantly, it does so at a lower cost, particularly when compared to methods that rely on iterative self-correction and feedback loops.
> >
> > We hope this explanation provides additional clarity and highlights the practical benefits of our approach.

---

> ### Author Response · Authors · 2024-12-02
> **Have we addressed your concerns ?**
>
> Dear Reviewer 4bru:
>
> Thank you again for your time and valuable feedback. As the rebuttal period is nearing its end, we hope our response has addressed your concerns. If so, we kindly request that you update your review and raise your score accordingly. Please let us know if you have any further questions or comments.
>
> Thank you!

---

### Author Response · Authors · 2024-11-27
**General Response**

**Response to Reviewer Feedback**

We greatly appreciate the constructive comments provided by the reviewers, which have significantly enhanced our manuscript. Below, we summarize the major revisions and responses to the key points raised:

**1. Expanded Experiments and New Baselines**
- Added three more NLP datasets for a total of eight, covering diverse domains.
- Evaluated one additional model: Added Meta-Llama 8B Instruct (open-source) alongside two existing closed-source LLMs, enhancing our comparative analysis.
- Introduced two new baselines, Self-Refine and Reflexion, demonstrating the superior performance of Retrospective Learning (RetroL).

**2. Validation of the CAI Ratio**
- Performed Pearson correlation analyses, confirming strong positive correlations between CAI ratios and accuracy, with significant p-values.
- Reported on CAI ratio disparities across models, underlining the lower accuracy of Meta-Llama 8B Instruct compared to others.

**3. Improvements in Presentation and Motivation**
- Clarified terminology, replacing "RL" with "RetroL" to prevent confusion with Reinforcement Learning.
- Enhanced the introduction to better articulate the challenges and necessity of a student model in scenarios where teacher model competency is low.
- Updated figures for clearer visual representation and alignment with the text.

**4. New Results and Observations**
- Presented new findings that show RetroL's significant improvements in annotation accuracy and CAI ratios across all datasets and models.
- Noted the framework's adaptability to both open-source and closed-source LLMs.

**5. Addressing Reviewer Comments**
- Provided further justifications for the student model, emphasizing its role in managing noisy teacher outputs.
- Clarified clustering operations and their relevance to our methodology, while outlining potential future work involving human experiments.

**Conclusion**
The rebuttal addresses all concerns raised about the reliability of the CAI ratio, the need for the student model, and comparisons with related methods. Our updates underscore the robustness and adaptability of the Retrospective Learning framework across a variety of datasets and LLM configurations.

---

### Meta-Review · Area_Chair_c5kY · 2024-12-20

**Metareview:**

The authors introduce "Retrospective Learning" to increase the accuracy of data generated for downstream tasks, e.g., to be used for evaluation. As far I as understand it, the problem being addressed here is basically trying to ensure "teacher competence" for new tasks which lack manual annotations. To this end, the authors introduce the notion of Consistent and Inconsistent (CAI) examples.

As pointed out by multiple reviewers (vKpA, 4bru), the motivation and framing is rather muddled in this paper. I read the abstract and introduction (even after being revised for clarity) and agree with this. The author response in OpenReview was actually more cogent, in my view. In any case, I think the work would need substantial revision from its current form to clarify the contributions on offer.

**Additional Comments On Reviewer Discussion:**

The authors did a nice job of being responsive to concerns. Reviewer d96A raised their score in response to additional experimental results, for example. However, the main issue with the work—its framing and clarity—remain an issue.

---

### Decision · Program_Chairs · 2025-01-22

Reject